# Overcoming platinum-resistant ovarian cancer targeting the activated JAK-STAT pathways via extracellular vesicles
Kazuhiro Suzuki [1], Akira Yokoi [1,2,3] ✉, Kosuke Yoshida [1,2], Hironori Suzuki[1,4], Masami Kitagawa[1], Eri Asano-Inami[1], Seiko Matsuo[1], Masato Yoshihara[1], Satoshi Tamauchi[1], Nobuhisa Yoshikawa [1], Kaoru Niimi[1], Tamotsu Sudo[5], Satoshi Yamaguchi[6], Yusuke Yamamoto [4] & Hiroaki Kajiyama [1]

Platinum-resistant ovarian cancer (PROC) is a clinically severe unresolved issue, and it remains unclearly defined by molecular biology. Extracellular vesicles (EVs) play an essential role in cell-to-cell communication in the tumor microenvironment. This study aimed to investigate the molecular mechanisms of PROC, focusing on the unique ascites environment of ovarian cancer. Multi-transcriptome analyses using clinical samples revealed that PROC exhibited an activated Janus kinase (JAK)/signal transducer and activator of transcription pathway with high JAK1 expression in cancer cells. Immunohistochemistry for patient tissues confirmed the negative association between JAK1 expression and platinum response. JAK inhibitors were effective in PROC cell lines and cell- and patient-derived xenograft models, as well as synergistic with platinum. Furthermore, small RNA sequencing indicated that activated peritoneal mesothelial cell-derived EVs enriched in miR135a-5p increased JAK expression and platinum resistance in cancer cells. Collectively, EVs in ascites regulated platinum sensitivity in ovarian cancer cells, and JAK targeting therapeutic strategy overcomes PROC.

Ovarian cancer is one of the leading causes of cancer death in women and the most fatal gynecologic malignancy, with high-grade serous ovarian cancer (HGSOC) as the most prevalent subtype[1]. Over 70% of patients are diagnosed at an advanced stage (International Federation of Gynecology and Obstetrics stages III and IV) with distant metastases[2]. Most patients with HGSOC experience relapse after initial treatment despite primary optimal debulking surgery and platinum-based chemotherapy[3]. These patients will progress from platinum-sensitive to platinum-resistant recurrence, which is the greatest challenge with limited treatment options, with increasingly shorter recurrence intervals.

Platinum-resistant ovarian cancer (PROC) remains unclearly defined by molecular biology. Clinically, we have categorized patients as PROC if they have relapsed <6 months after their last platinum-based chemotherapy (treatment-free interval of platinum [TFIp]) for the past 30 years[4–6]. Patients clinically categorized as PROC have not received platinum-based chemotherapy. Recently, clinical trials have revealed that some patients with

PROC may benefit from platinum-based chemotherapy[7]. Hence, the definition of PROC has been questioned and debated[7]. However, the prognosis for disease categorized as PROC is extremely poor, with a progression-free survival of approximately 6 months[8]. Several mechanisms of platinum resistance have been reported in molecular biology, such as decreased intracellular accumulation[9], tumor microenvironment[9], DNA repair-related pathways[10], and cancer stem cells[11]. Research has been conducted at the cellular and genetic levels but results remain unclear.

Most of the patients with advanced ovarian cancer present with cancerous ascites when diagnosed as well as platinum resistant[12]. Cancerous ascites contain a mixture of diverse cells, including cancer cells, and their interaction causes peritoneal dissemination[13]. Cells in ascites release extracellular vesicles (EVs), and EVs contain bioactive molecules[14]. EVs in ascites play a role in cell-to-cell communication, affecting the peritoneal dissemination microenvironment in addition to direct cell-to-cell actions[15]. The function of EVs in ascites in PROCs remains unknown.

[1]Department of Obstetrics and Gynecology, Nagoya University Graduate School of Medicine, Nagoya, Aichi, Japan. [2]Institute for Advanced Research, Nagoya University, Furo-cho, Chikusa-ku, Nagoya, Aichi, Japan. [3]Japan Science and Technology Agency, Fusion Oriented Research for Disruptive Science and Technology, Saitama, Japan. [4]Laboratory of Integrative Oncology, National Cancer Center Research Institute, Tokyo, Japan. [5]Department of Genomic Medicine, Fujita Health University School of Medicine, Toyoake, Aichi, Japan. [6]Department of Gynecologic Oncology, Hyogo Cancer Center, Akashi, Hyogo, Japan. ✉e-mail: ayokoi@med.nagoya-u.ac.jp

This study indicated that bulk RNA sequencing, spatial transcriptome analysis, and small RNA sequencing of clinical samples from chemo-naive patients with HGSOC revealed that activated mesothelial cell-derived miR-135a-5p-enriched EVs in ascites affect cancer cells and activate the Janus kinase/signal transducer and activator of transcription (JAK-STAT) pathway, causing platinum resistance in cancer cells. Subsequent analysis revealed that JAK inhibitors have antitumor effects for platinum resistance and synergistic effects with platinum, and normal mesothelial cell-derived EVs unlocked platinum resistance in cancer cells, indicating phenotypic plasticity.

## Results

### JAK-STAT pathway in cancer cells is associated with PROC

We performed bulk RNA sequencing of 20 HGSOC cases to determine the pathways involved in platinum resistance. Tissue samples included chemotherapy-naive primary ovarian cancer tissue, collected during the initial surgery, ten PROC, and ten platinum-sensitive cases, respectively (Fig. 1a,b). Supplementary Table 1 shows the clinical information of the patients. The heatmap and principal component analysis (PCA) revealed that the gene expression profile of PROC was different from that of the platinum-sensitive groups (Fig. 1c and Supplementary Fig. 1a). Multivariate analysis of 12 selected cases with differences in PCA to better reveal differences between the PROC and the platinum-sensitive groups exhibited 480 upregulated genes in the PROC group and 485 in the platinum-sensitive group, based on a cut-off of |log2FC| of >1.1 and an adjusted $P$-value of <0.05 (Fig. 1d and Supplementary Fig. 1b). Pathway analysis was conducted using Ingenuity pathway analysis (IPA) software, which revealed that several pathways associated with the JAK-STAT pathway were significantly upregulated, to evaluate the putative function of the 965 differentially expressed genes (Fig. 1e). In particular, these pathways included the role of JAK1, JAK2, and TYK2 in interferon signaling ($P = 4.79E-06$), interleukin (IL)-15 production ($P = 7.08E-06$), the role of PKR in interferon induction and antiviral response ($P = 9.77E-05$), JAK/STAT signaling ($P = 3.80E-04$), and interferon signaling ($P = 4.57E-04$). High *JAK-STAT* family gene expression was shown in PROC tissues of our clinical samples, and the same results were validated in the Gene Expression Omnibus database (Fig. 1f, g and Supplementary Fig. 1c, d). The Cancer Genome Atlas (TCGA) data revealed that *JAK-STAT* family genes are more highly expressed in normal ovaries than in cancer tissues, but a greater correlation was found between *JAK* and *STAT* family gene expression in cancer tissues, indicating that the JAK-STAT pathway is responsive in cancer (Fig. 1h and Supplementary Fig. 1e, f). Furthermore, high *JAK-STAT* gene expression in ovarian cancer tissue is related to poor prognosis (Fig. 1i and Supplementary Fig. 1g). Therefore, we conducted spatial transcriptomics analysis to evaluate the expression of *JAK-STAT* family genes in cancer cells within PROC tissues (Fig. 1b). Tumor samples from PROC (HGS-76R and HGS-36R) and platinum-sensitive (HGS-72S and HGS-36S) cases underwent Visium spatial gene expression analysis (Fig. 1j). High *JAK-STAT* family gene expression was confirmed in cancer cell annotated regions of PROC tissues compared to cancer cell annotated regions of platinum-sensitive tissues (Fig. 1k–l and Supplementary Fig. 2a, b). We analyzed the single-cell RNA sequencing dataset to validate the high expression of *JAK-STAT* family genes in cancer cells within PROC tissues[16]. *JAK1* expression was higher in cancer cells within PROC tissues where data on its expression levels were available for analysis (Supplementary Fig. 2c–e). Thus, the *JAK* family genes are highly expressed in cancer cells within PROC tissues, and the JAK-STAT pathway is upregulated in PROC tissues.

### Clinical relevance of JAK-STAT in HGSOC

Immunohistochemistry for JAK1, STAT1, and STAT3 was performed on samples of patients with ovarian cancer to confirm the correlation between JAK-STAT family expression and platinum treatment response. JAK1, STAT1, and STAT3 expressions were stronger in PROC than in the platinum-sensitive group of primary chemotherapy-naive cancer tissues (Fig. 2a, b). Additionally, we revealed strong expression of JAK1 and STAT1

in PROC surgical samples during recurrence (Fig. 2c, d). We then confirmed changes in the JAK-STAT family expression over the treatment course. Weak JAK1 and STAT1 expression persisted in HGS-85S cases that remained platinum-sensitive at recurrence (Fig. 2e). In contrast, the HGS-04R case was a PROC intraoperatively during recurrence and had stronger JAK1, STAT1, and STAT3 expressions (Fig. 2f).

### Effects of JAK inhibition on PROC cell lines

We examined the expression levels of JAK-STAT in paired ovarian cancer cell lines, including parental and platinum-resistant lines. Several *JAKs* were found to be highly expressed at the mRNA level in resistant cell lines (Fig. 3a and Supplementary Fig. 3a). At the protein level, high expression of JAK1 as well as phosphorylation of JAK2 and JAK3 were observed in A2780cis cells, while phosphorylation of JAK1 was confirmed in PEO1-CDDP cells. (Supplementary Fig. 3b). We first performed gene silencing experiments using siRNAs to evaluate cell line proliferation by *JAK* family inhibition (Fig. 3b and Supplementary Fig. 3c). *JAK1*, *JAK2*, *JAK3*, and *TYK2* knockdown by siRNA suppressed A2780cis and PEO1-CDDP cell proliferation without affecting cell morphology (Fig. 3c, d). Peficitinib inhibited cell proliferation of A2780cis to confirm the inhibitory effect of JAK inhibitors on cell proliferation (Fig. 3e). We revealed that JAK inhibitors exhibit equivalent cytotoxic effects on PROC and parental cell lines (Fig. 3f).

### In vivo efficacy of JAK inhibition

We studied two types of ovarian cancer model mice, including cell line-derived xenograft (CDX) and patient-derived xenograft (PDX), to evaluate the clinical potential value of JAK inhibitors. The CDX mouse models were generated by intraperitoneal injection of luciferase-transfected A2780 and A2780cis, and tumor formation was monitored over time. We assessed the anticancer effect of intraperitoneal peficitinib and revealed that it inhibited tumor growth in A2780-CDX and A2780cis-CDX (Fig. 4a–c and Supplementary Fig. 4a–c). PDX model mice were generated from tumors of patients with PROC (HGS-PDXR) (Fig. 4d). These HGS-PDXR mouse tumors demonstrated high *JAK-STAT* expression compared to PDX model mice generated from patients with platinum-sensitive (HGS-PDXS) (Fig. 4e). Peficitinib demonstrated an anticancer effect in the HGS-PDXR mice that formed subcutaneous tumors (Fig. 4f, g). The anticancer effect was supposed to be the result of apoptosis induced by Peficitinib, according to immunostaining of the excised tumor, which was positive for cleaved caspase-3 and negative for RIPK3 (Supplementary Fig. 4d). No significant difference in the weight of the mice was found between groups (Supplementary Fig. 4e–g).

### Identification of the miRNA profiles of PROC

We investigated JAK-STAT pathway activation in PROC within cancer cells from the perspective of the intra-abdominal environment where ovarian cancer develops. Analysis was conducted on ascites and tumor samples of HGSOC to determine miRNAs involved in PROC. Comprehensive miRNA sequencing was conducted using ascites small EVs (ascites-EVs) and cancer tissues (Fig. 5a). Ascites-EVs were isolated following the protocol and characterized by nanoparticle tracking analysis, transmission electron microscopy, and western blotting (Fig. 5b and Supplementary Fig. 5a). Heatmap revealed that miRNA profiles in ascites-EVs exhibited no specific clusters in the resistant and susceptible groups (Fig. 5c). However, the PCA indicated a certain trend of expression in the platinum-resistant and platinum-sensitive groups (Fig. 5d). The miRNA expression profile in normal ascites-EVs was different from that of cancer ascites (Supplementary Fig. 5b). Heat map and PCA revealed clear differences in expression trends between the platinum-resistant and platinum-sensitive groups regarding the miRNA profiles in the tissues (Fig. 5e, f). Volcano plots revealed significantly higher 12 miRNA expression in ascites-EVs and 35 miRNAs in tissues (Fig. 5c, e). Among the miRNAs with high expression in the resistant group, miR-135a-5p

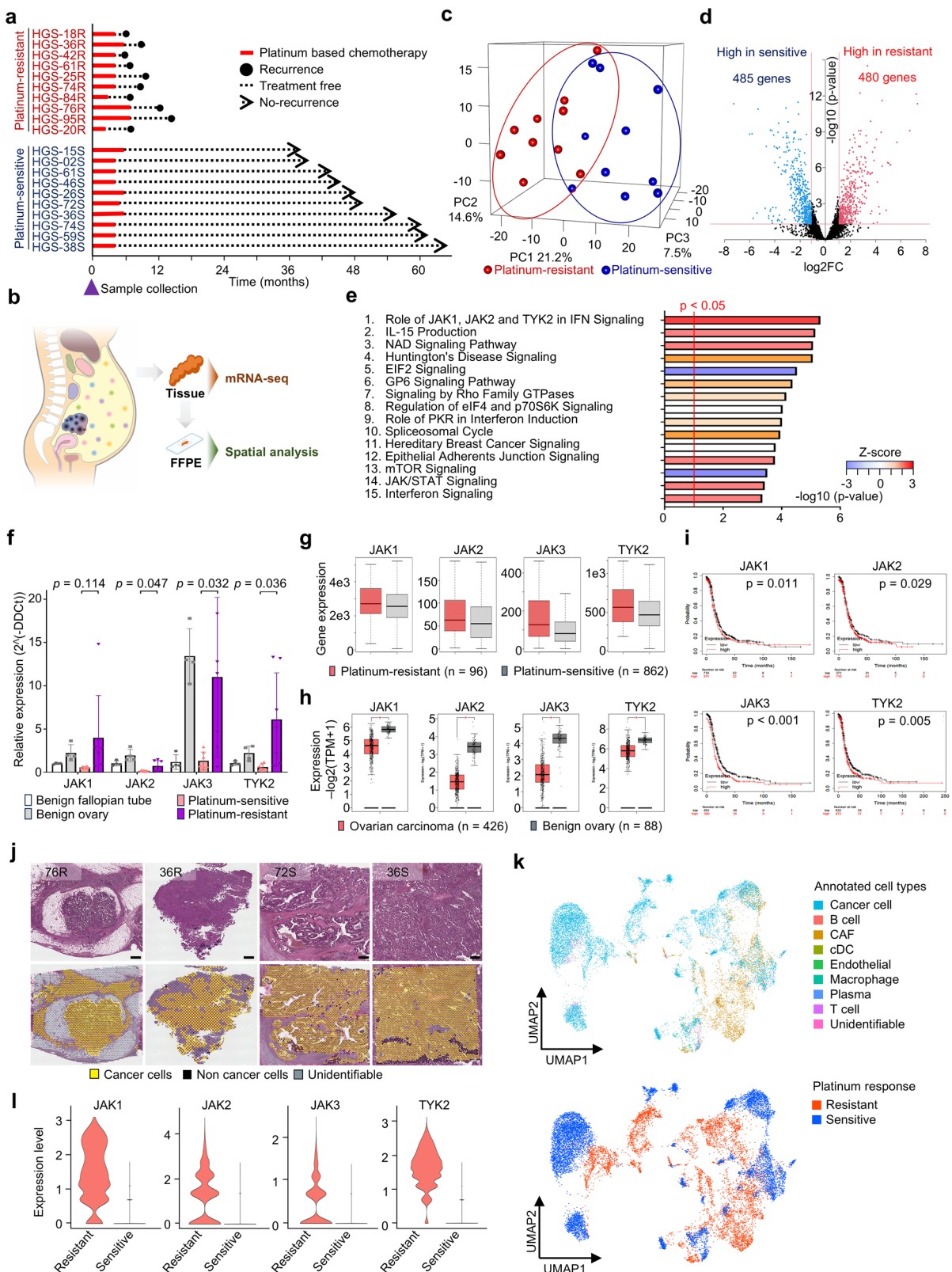

and miR-221-5p were predominantly highly expressed in ascites-EVs and tissues (Fig. 5g). Both miRNA types demonstrated a positive correlation between miRNA expression in ascites-EV and tissue ($r^2 = 0.938$ for miR-135a-5p and $r^2 = 0.942$ for miR-221-5p, respectively) (Fig. 5h). Although there was no statistically significant difference, high expression of both miRNA types in ovarian cancer tissues tended to be associated with

shorter overall survival based on Kaplan–Meier curves (Supplementary Fig. 5c). We assessed the cisplatin sensitivity using miRNA-transfected ovarian cancer cell lines. The 50% inhibitory concentration (IC50) of cisplatin in KURAMOCHI cells transfected with two miRNAs was 2.4-fold (miR-135a-5p) and 2.1-fold (miR-221-5p) that of negative control (NC) mimics (Fig. 5i). Additionally, the IC50 of cisplatin increased by the

**Fig. 1 | Identification of JAK-STAT signaling pathways in PROC. a** Timeline of 20 patients with HGSOC whose tumors were utilized for bulk mRNA sequencing, spatial transcriptome analysis, and miRNA sequencing, including 44 unique platinum chemotherapy-naive samples. **b** Study design for transcriptome analysis based on initial surgical tissue samples of HGSOC. **c** Principal component analysis from bulk mRNA sequencing, including ten platinum-resistant and ten platinum-sensitive samples. **d** Volcano plot demonstrating significant differentially expressed genes between platinum-resistant and platinum-sensitive, based on a cut-off of |log2FC| > 1.1 and an adjusted *P*-value of <0.05. Adjusted *P*-values for each gene were calculated using the Wald test in DESeq2. **e** Top 15 significantly upregulated pathways in PROC based on IPA for the 964 significant differentially expressed genes. **f** *JAK* family expression in tissues by qRT-PCR, including 4 benign ovarian tubes, four benign ovaries, seven platinum-sensitive HGSOC, and seven resistant HGSOC tissue samples. **g** *JAK* family expression between platinum-resistant and platinum-sensitive HGSOC in 11 GEO and TCGA datasets using ROC Plotter.

**h** *JAK* family expression between ovarian carcinoma and benign ovary using the GEPIA2 database. **i** Kaplan–Meier plots of disease-free survival following *JAK* family expression in HGSOC using the Kaplan–Meier Plotter. *P*-values were calculated using the log-rank test. **j** H&E staining of high-grade serous ovarian cancer (HGSOC) tissue sections from platinum-resistant cases (HGS-76R and HGS-36R) and platinum-sensitive cases (HGS-72S and HGS-36S). Supplementary Table 1 shows the clinical information of the patients. Scale bars indicate 500 μm. Visium spatial transcriptomics spots on H&E stained sections were overlaid with spatial feature spots of ovarian cancer cells, noncancer cells, and unidentified. SpaCET annotated spot types with a gene pattern dictionary of copy number and expression changes. **k** UMAP plots of Visium spatial transcriptomics spots. Spatial transcriptomics spots were classified into nine clusters by annotation using SpaCET and into two clusters based on platinum response. CAF: cancer-associated fibroblasts; cDC: conventional dendritic cell. **l** Violin plot of *JAK* family expression in the cancer cell subpopulations.

transfection of miR-135a-5p and miR-221-5p in other ovarian cancer cell lines (Fig. 5i and Supplementary Fig. 5d, e).

### Mesothelial cells in ascites enriched for miR-135a-5p

We first focused on the constituent cells in the ascites to identify which cells in the ascites were rich in the two candidate miRNAs. The coexistence of cancer cells and mesothelial cells was observed in formalin-fixed paraffin-embedded blocks prepared from malignant ascites of patients with HGSOC (Fig. 6a, b). The coexistence of cancer cells and mesothelial cells in the ascites of patients with HGSOC was validated based on single-cell RNA sequencing datasets (Fig. 6c)[16]. *JAK-STAT* expressions were increased by transfection of miR-135a-5p and miR-221-5p in ovarian cancer cell lines (Fig. 6d and Supplementary Fig. 6a, b). Exposure of ovarian cancer cell lines to TGFβ, which is one of the ascites cytokines involved in resistance[17–20], did not increase *JAK-STAT* expression or alter cisplatin sensitivity (Supplementary Fig. 6c–e). However, TGFβ treatment of ascitic constituent cells significantly upregulated two miRNA species in MTK cells, a type of peritoneal mesothelial cell, with a particularly marked increase in miR-135a-5p expression (Fig. 6e).

### Effect of mesothelial cell-derived EVs in ascites on ovarian cancer cell

EVs were isolated from MTK cell culture medium and EVs were characterized to confirm whether EVs derived from activated MTK cells by TGFβ are enrich in miR-135a-5p (Fig. 6f and Supplementary Fig. 6f, g). We revealed that miR-135a-5p was enriched in EVs derived from activated MTK cells (Fig. 6g). To further confirm whether the enriched miRNAs are encapsulated within the EVs or associated with their surface, mesothelial cell-derived EVs were treated with RNase and their expression was measured by miRNA quantitative PCR. There were no differences in the expression of miR-135a-5p in EVs between RNase-treated and untreated groups, indicating that the enriched miRNAs are contained within the EVs (Fig. 6h and Supplementary Fig. 6h). Next, we investigated changes in *JAK-STAT* expression and cisplatin sensitivity on KURAMOCHI with the EVs derived from activated MTK cell (Supplementary Fig. 6i). KURAMOCHI, in which isolated EVs were confirmed to uptake, demonstrated increased *JAK1*, *JAK2*, and *TYK2* expression and decreased cisplatin sensitivity (Fig. 6i–k and Supplementary Fig. 6j). Similarly, cisplatin sensitivity of KURAMOCHI was decreased by miR-135a-5p uptake-enriched EVs derived from MTK cells transfected with miR-135a-5p mimic (Fig. 6l and Supplementary Fig. 6k, l). Conversely, cisplatin sensitivity was increased by siRNA knockdown of *JAK1*, *JAK2*, *JAK3*, and *TYK2* in A2780cis and PEO1-CDDP (Supplementary Fig. 7a). We revealed that the combination of peficitinib and cisplatin inhibited cell proliferation of A2780cis more than cisplatin alone and exhibited a synergistic effect (Supplementary Fig. 7b, c). We investigated changes in *JAK-STAT* expression and cisplatin sensitivity on A2780cis by the EVs derived from non-activated MTK cells (Supplementary Fig. 8a).

A2780cis, in which isolated EVs were confirmed to uptake, exhibited decreased *JAK1*, *JAK2*, and *JAK3* expression and increased cisplatin sensitivity (Fig. 6m, n and Supplementary Fig. 8b, c).

## Discussion

This study revealed that the expression of miR-135a-5p in both ascites-EVs and cancer tissues was upregulated in platinum resistance, indicating activation of the JAK-STAT pathway in PROC. JAK inhibitors were effective against PROC cells and had synergistic effects with platinum drugs. Subsequent analysis revealed that EVs derived from activated mesothelial cells are enriched in miR-135a-5p, and these EVs enable cancer cells to acquire platinum resistance. Furthermore, EVs released from normal mesothelial cells changed the phenotype of cancer cells from platinum resistance to sensitivity (Supplementary Fig. 8,d). Thus, we have discovered that EVs in ascites mediate cancer cellular plasticity and alter platinum sensitivity through cell-to-cell communication.

Treatment resistance may be inherent or acquired during therapy by different adaptive reactions[21]. The strength of this study is the results from the analysis of the pure-resistant clinical samples that did not respond to initial platinum treatment. Our transcriptome analysis may have revealed the status of intrinsic resistance, but the acquired resistance is hypothesized to be analyzed only for the status remaining after selection[21,22]. Additionally, the immunohistochemistry results demonstrate an increased JAK-STAT positive rate in recurrent PROC, indicating that JAK-STAT plays a pivotal role in PROC.

Ovarian cancer has a plasticity characterized by heterogeneity at the molecular[23] and cellular levels[24]. Factors that contribute to cancer cell resistance include newly acquired genetic mutations and transformation due to phenotypic plasticity. JAK-STAT pathway-mediated plasticity of cancer cell resistance to therapy has been previously reported in urologic cancers[25,26]. Single-cell RNA analysis of ascites indicated that the JAK-STAT pathway is a therapeutic target in ovarian cancer[27]. Single-cell analysis helps us to determine which cells undergo which changes, but determining the underlying causes of these changes remains challenging. Therefore, the present study revealed that the mesothelial cell-derived EVs in the ascites mediated JAK-STAT pathway activation in cancer cells, causing therapeutic resistance, thereby providing further information on the intraperitoneal cancer microenvironment and therapeutic targets.

Numerous clinical trials have been conducted for platinum resistance in ovarian cancer, but have not exhibited marked improvement in prognosis. Our identified JAK inhibitor efficacy has already been investigated in clinical trials in the ovarian cancer front-line setting[28]. This clinical trial[28] indicates that JAK inhibitors may have salvaged the initial state of platinum resistance that was initially resistant to initial treatment, as our preclinical study revealed. JAK inhibitors can be effective in platinum retreatment due to the plasticity of JAK-STAT pathway regulation to unlock platinum resistance. Accordingly, JAK inhibitors represent a potential salvage strategy for patients with ovarian cancer with poor prognosis and should be translated into clinical applications as soon as possible.

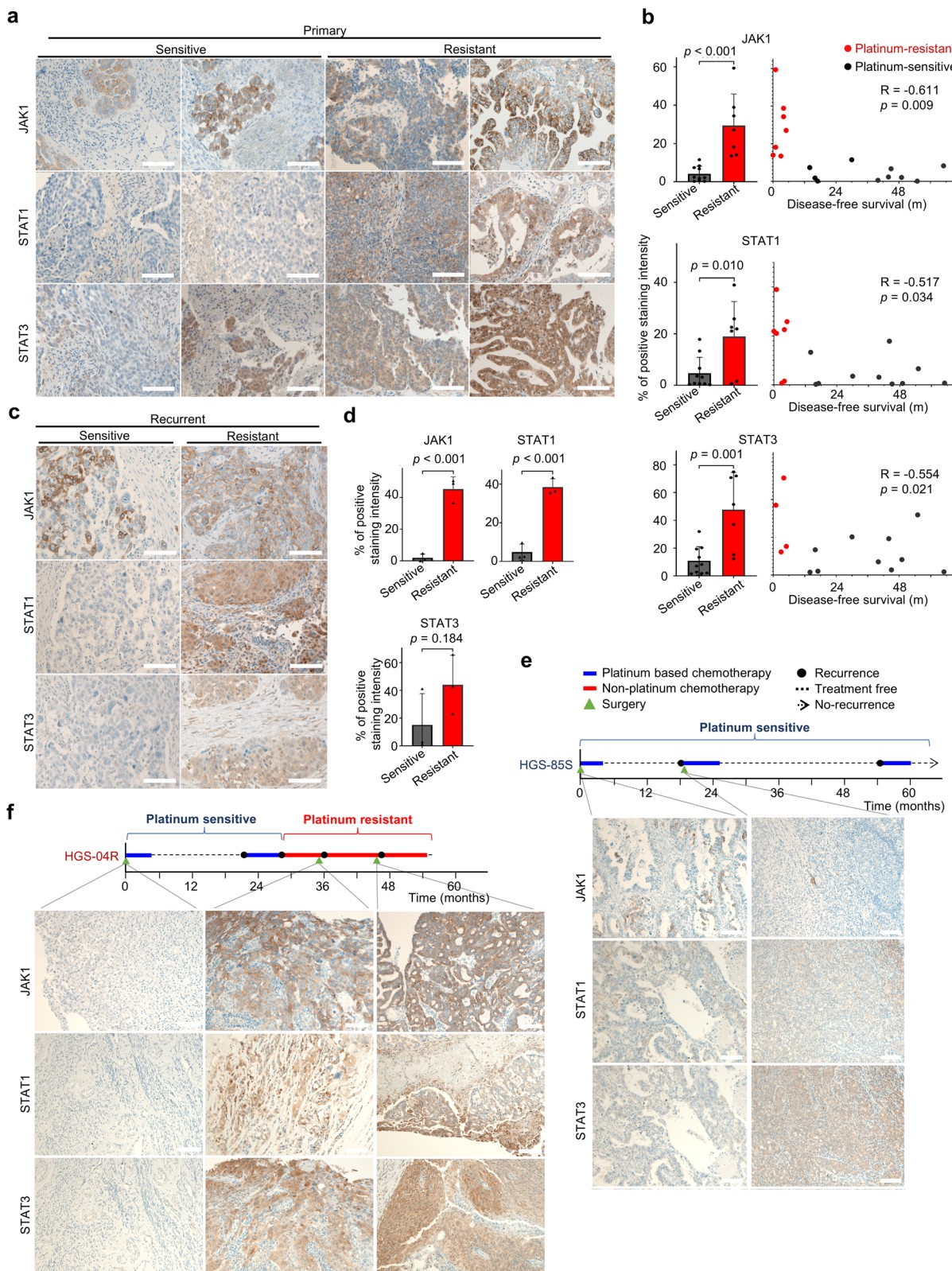

The ovarian cancer-specific pathogenesis of cancerous ascites and peritoneal dissemination has prompted various studies of the cells and humoral factors present in ascites[16,29]. The abdominal cavity is a structure that is entirely covered by a monolayer of peritoneal mesothelial cells. Therefore, the ascites contain EVs derived from peritoneal mesothelial cells. The present study revealed that the phenotypic plasticity of cancer cells is mediated by EVs derived from peritoneal mesothelial cells. The typical treatment history of ovarian cancer is that the peritoneal dissemination disappears after successful initial treatment, but recurrence repeatedly occurs and the peritoneal dissemination remains and becomes platinum resistant. Platinum sensitivity preservation for some time may indicate that treatment response is due to the phenotypic plasticity of ovarian cancer cells

**Fig. 2 | JAK-STAT expression in primary and recurrent PROC tissues.**
**a** Representative images of immunohistochemistry of JAK1, STAT1, and STAT3 in primary chemotherapy-naive HGSOC tissue. Scale bars indicate 100 μm. **b** Protein expression of JAK1, STAT1, and STAT3 in chemotherapy-naive HGSOC tissue by immunohistochemistry based on the number of positive areas per visual field. Welch's *t*test was used for comparison and *p*-values are shown in the graph. R-values were calculated from Spearman's correlations between % positive areas and disease-free survival. (*n* = 7, resistant; *n* = 10, sensitive). **c** Representative images of immunohistochemistry for JAK1, STAT1, and STAT3 in recurrent HGSOC tissue. Scale

bars indicate 100 μm. **d** Protein expression of JAK1, STAT1, and STAT3 in recurrent HGSOC tissue by immunohistochemistry based on the number of positive areas per visual field. Welch's *t*-test was used for comparison and *p*-values are shown in the graph. R-values were calculated from Spearman's correlations between % positive areas and disease-free survival. (*n* = 3, resistant; *n* = 3, sensitive). **e** Time course of immunohistochemistry of cancer tissue at surgery in the same patient with persistent sensitivity. Scale bars indicate 100 μm. **f** Time course of immunohistochemistry of cancer tissue at surgery in the same patient with resistance at the second surgery. Scale bars indicate 100 μm.

to platinum sensitivity, as peritoneal dissemination disappears with initial treatment and normalized peritoneal mesothelial cells are preserved.

EVs are abundant in ovarian cancer ascites and affect tumorigenesis, metastasis, and chemoresistance[15,30]. Much attention has been focused on ovarian cancer cell-derived EVs, and we have now determined a function for peritoneal mesothelial cell-derived EVs. The cellular component of ascites includes hematopoietic cells, immune cells, and other cells, in addition to cancer cells and mesothelial cells[16,27,31]. Our study has limitations. First, we did not conduct a comprehensive analysis of the functions of EVs derived from all cells in the ascites. EVs released by various unexamined cell types may play a role in regulating the intraperitoneal cancer microenvironment and could contribute to platinum resistance. Ideally, we would formally establish the mechanisms of EV-mediated delivery, but this remains a significant challenge, representing a limitation of our study. Second, in our in vitro experiments, not all JAK family members exhibited similar expression changes. This may be due to the limitations of cell lines, which reflect uniform, monoclonal changes. Additionally, JAK-STAT signaling can be activated through JAK phosphorylation as mono- or heterodimers, even if some members are not highly expressed. Third, the mechanism by which EV-miR135a-5p improved JAK expression was not elucidated in detail. EV-miR-135, which is the ascites-EV focused on in this study, may serve as a direct target for JAK3, which is a typical mechanism by which miRNAs regulate gene expression by repressing translation or directing sequence-specific degradation of complementary mRNAs[32–34]. However, miR-135a-5p rather works in the direction of improving JAK3 expression, which is an emerging function of miRNAs, considering that they activate gene expression by interacting with complementary regions observed in the promoter and coding region[35,36].

Collectively, our results reveal an important role in cancer cell plasticity played by peritoneal mesothelial cell-derived EVs in ascites. We have revealed that JAK inhibitors are effective in platinum-resistant cancer cells with high JAK-STAT expression, and that high miR135 expression in ascites is a contributing factor to these results. Moreover, we found that platinum resistance is modulated by normal peritoneal mesothelial cells that release EVs, suggesting a potential for therapeutic strategies targeting cancer cell plasticity.

## Materials and methods
### Patients
This study used archival fresh-frozen tumor samples and ascites that are stored at the Hyogo Cancer Center Biobank (Hyogo, Japan). Ovarian cancer tissue was collected during primary surgery before any treatment, ascites were collected at the beginning of surgery, and relevant clinical data were obtained. This study included 20 patients diagnosed with stages III–IV HGSOC, including ten patients with an initial treatment-free interval of platinum (TFIp) of <6 months (platinum-resistant group) and ten patients with an initial TFIp of ≥36 months (platinum-sensitive group). Normal ovarian, fallopian tube, and ascites samples were collected intraoperatively at Nagoya University Hospital, Nagoya, Japan (*n* = 12) from 2019 to 2022. The Institutional Review Board of Hyogo Cancer Center (approval No. 2021–911) and Nagoya University Hospital (approval No. 2017–0053) approved the study protocol. Written informed consent was obtained from all patients. This study was conducted under the Declaration of Helsinki.

### RNA extraction and transcriptome analysis
The miRNeasy Mini Kit (QIAGEN, Hilden, Germany) was used to extract total RNA, following the manufacturer's instructions, and a NanoDrop ND-1000 spectrophotometer (Thermo Fisher Scientific) was utilized to measure total RNA concentration. Novogene, Inc. (Tokyo, Japan) conducted RNA sequencing. Kallisto quantified the expression levels for each gene. We prepared small RNA libraries with the NEBNext Multiplex Small RNA Library Prep Set for Illumina (New England Biolabs, Ipswich, MA) and added index codes to attribute sequences to each sample. PCR products were then purified using the QIAquick PCR Purification Kit (Qiagen) and 6% TBE gel (120 V, 60 min). Furthermore, DNA fragments corresponding to 140–160 bp (the length of small non-coding RNA plus the 3' and 5' adapters) were recovered, and the Qubit dsDNA HS Assay Kit and a Qubit 2.0 Fluorometer (Life Technologies, Carlsbad, CA) were used to measure the complementary DNA concentration. Finally, single-end reads were performed on the Illumina MiSeq (Illumina, San Diego, CA). The data were then summarized using the tximport package (version 1.18.0) of R software (version 4.0.3) and RStudio (RStudio, Boston, MA), and scaled TPM counts were utilized for further analysis. Genes with low read coverage (maximum read count: <10 reads) were excluded, and differentially expressed genes (DEGs, |log2FC = >1) were used for the heatmap and PCA. The heatmap.2 function of the gplots package (version 3.1.0) was utilized after converting the data to base 10 logarithms and z-scores. PCA was conducted with the prcomp and plot3d functions of the rgl package (version 0.100.54). The adjusted *p*-values for each gene were calculated with the Wald test in DESeq2 (version 1.30.0), and pathway analysis was conducted using the IPA (QIAGEN) software.

### Isolation of EVs from ascites samples
Approximately 1 mL of each ascites was centrifuged at 10,000 g for 40 min at 4 °C in the Kubota Model 3520 ultra-centrifuge. The supernatant was filtered using a 0.22 m filter (Millex-GV 33 mm, Millipore), and then ultra-centrifuged at 110,000 g for 70 min for 4 °C using a TLA55 rotor (Beckman Colter Inc., USA). The pellet was washed with phosphate-buffered saline (PBS), ultracentrifuged under the same conditions, and resuspended in PBS to extract small EVs (sEVs). EVs are isolated by ultracentrifugation and divided into large EVs and sEVs. However, we only use sEVs in our analysis, and in this article, EVs are used to mean sEVs.

### Spatial transcriptomics
We used the Visium CytAssist Spatial Gene Expression for formalin-fixed paraffin-embedded (FFPE) (10x Genomics, USA) for spatial transcriptomics. The whole RNA transcriptome of cells in each spot of specialized slides was obtained from FFPE tissue sections in this analysis. Each spot contained approximately 10–20 cells, and the RNA transcriptome of these cells revealed the features of RNA expression in each spot. CyberomiX (Kyoto, Japan) conducted hematoxylin and eosin (H&E) staining, RNA library preparation, and transcriptome sequencing as follows. Sections were H&E stained, imaged, and de-cover-sipped, followed by H&E de-staining and de-crosslinking. A Visium CytAssist instrument was used to process glass slides with tissue sections to transfer analytes to the Visium CytAssist Spatial Gene Expression slide with a 6.5 × 6.5 mm capture area. Probe extension and library construction steps followed the standard Visium for the FFPE workflow. Libraries were sequenced using DNBSEQ-G400

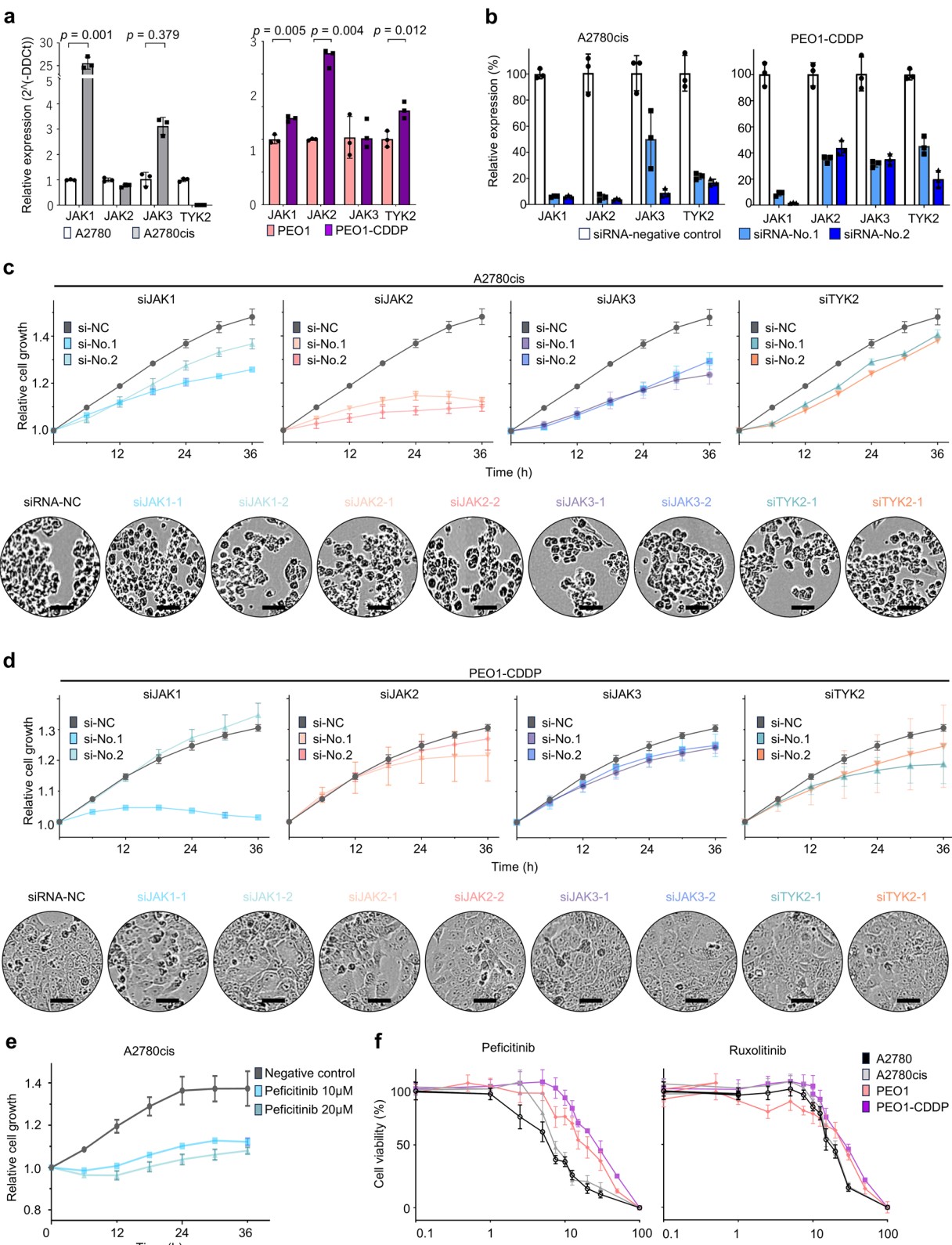

**Fig. 3 | Effects of JAK inhibition in PROC cell lines. a** Relative *JAK* family mRNA expression in paired parental and PROC cell lines. GAPDH was utilized as a reference gene to normalize expression. **b** *JAK1*, *JAK2*, *JAK3*, or *TYK2* mRNA suppression validation after transfection with 3 nmol/L of siRNA for siRNA-No.1 or siRNA-No.2 for 24 h. **c** Proliferation of siRNA-transfected A2780cis and the representative images of siRNA-transfected cells. Cell viability was measured every 6 h after transfection of 48 h. The experiment was conducted in triplicate. Scale bars indicate 50 μm. **d** Proliferation of siRNA-transfected PEO1-CDDP and the representative images of siRNA-transfected cells. Cell viability was measured every 6 h after transfection of 48 h. The experiment was conducted in triplicate. Scale bars indicate 50 μm. **e** A2780cis proliferation in peficitinib treatment. Cell viability was measured every 6 h. The experiment was conducted in triplicate. **f** Effect of JAK inhibitors: peficitinib and ruxolitinib. Cells were treated with peficitinib or ruxolitinib for 72 h. Black, gray, pink, and purple colors represent A2780, A2780cis, PEO1, and PEO1-CDDP, respectively.

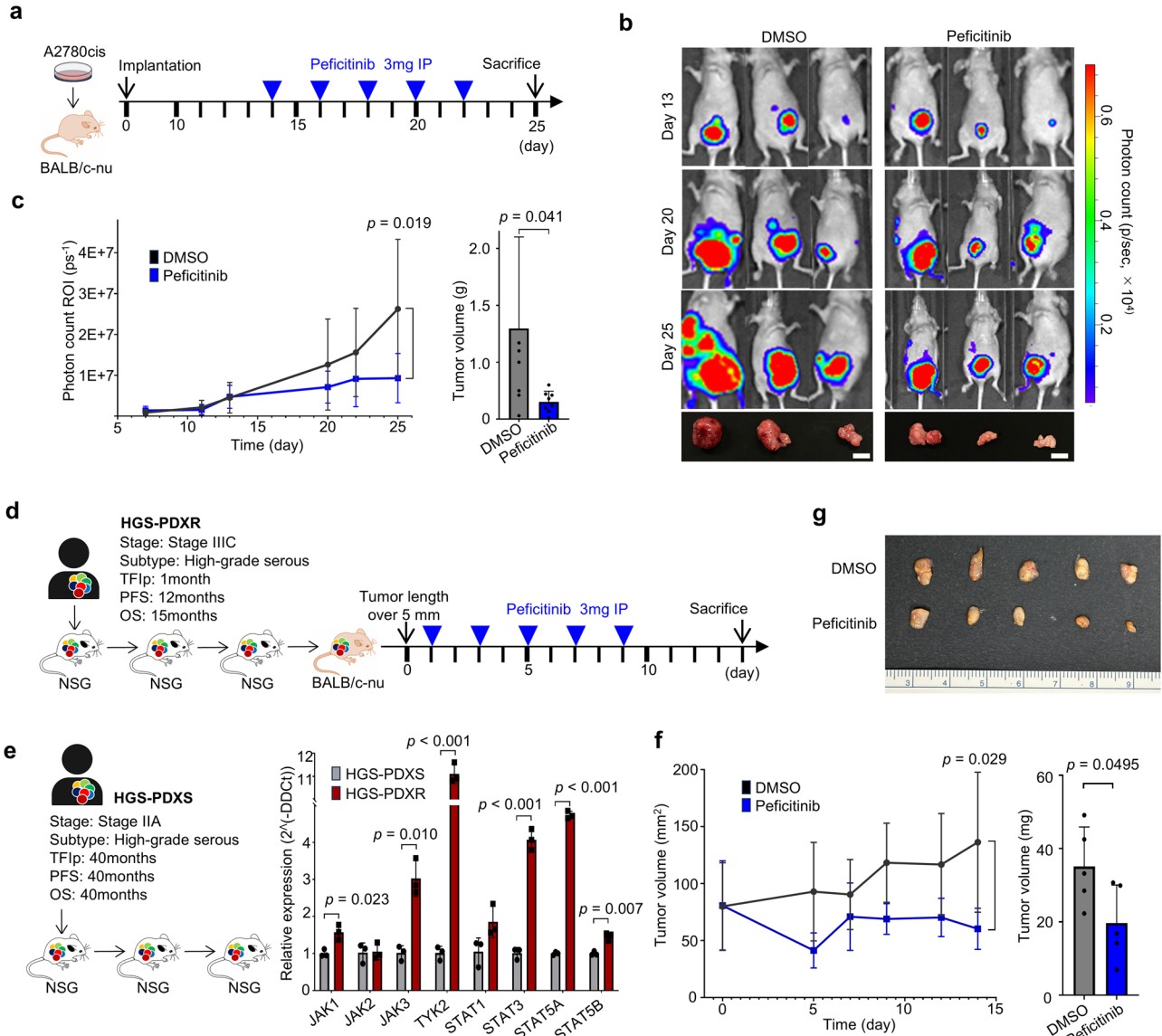

**Fig. 4 | Antitumor effects of JAK inhibitors in the CDX and PDX models.**
**a** Schema of the treatment schedule in A2780cis platinum-resistant cell line-derived xenograft model. Peficitinib or DMSO was intraperitoneally administered once every 2 days for a total of five injections. **b** Representative images of IVIS images and resected tumors. Photon values were evaluated once a week with an IVIS imaging system. Scale bars indicate 10 mm. **c** Quantitative analysis of tumor progression as bioluminescence images and resected tumors. Relative photon counts and mean tumor volume of A2780cis-bearing mice treated with either DMSO or peficitinib. Relative photon count on day 25 and resected tumor volume were compared using Welch's *t*-test. (peficitinib, *n* = 8 mice; DMSO, *n* = 8 mice). **d** Schema of the treatment schedule in patients with PROC (HGS-PDXR)-derived xenograft model.

The clinical background of the patient is as described. Peficitinib or DMSO was intraperitoneally administered once every 2 days for a total of five injections. **e** Scheme for patient-derived xenograft model of platinum-sensitive HGSOC (HGS-PDXS). The clinical background of the patient is as described. Relative *JAK-STAT* family mRNA expression in tumors of paired HGS-PDXS and HGS-PDXR mice. **f, g** Quantitative analysis of tumor progression as estimated and resected tumor. Estimated tumor volume of HGS-PDXS tumor-bearing mice treated with either DMSO or peficitinib. Estimated tumor volume on day 14 and resected tumor volume were compared using Welch's *t*-test. (peficitinib, *n* = 5 mice; DMSO, *n* = 5 mice).

sequencer (BGI) (read 1: 100 bp, read 2: 100 bp). Space Ranger (2.0.1, 10× Genomics) processed the Visium sequencing data. The output count matrix and image data were normalized, quality controlled, batch effect corrected, and dimension reduced with the R package Seurat (v5.0.0). Spots with an nCount of 0 were removed, and normalization was conducted with the function SCTransform in Seurat. Batch effect[37] correction, integration by harmony[38], and reduction and clustering were performed following Seurat vignettes (version 5.0.0). *P*-values are adjusted using the Benjamini-Hochberg correction for multiple tests. The analysis data were exported from the Loupe browser, and a volcano plot was plotted by ggplot2 (3.4.4) in R.

## Cell lines
The following seven cell lines were utilized as human ovarian cancer models: KURAMOCHI, A2780, A2780cis, PEO1, PEO1-CDDP, CAOV3, and SKOV3. The Japanese Collection of Research Bioresources Cell Bank provided KURAMOCHI cells. A2780, A2780cis, PEO1, and PEO1-CDDP were purchased from the European Collection of Authenticated Cell Cultures. CAOV3 and SKOV3 were purchased from the American Type Culture Collection. Cell lines were maintained by culturing in their optimal medium and conditions following the suppliers' recommendations. Four noncancer cell lines included MTK, Met5A, and HFFF2. MTK cells, derived from human greater omental

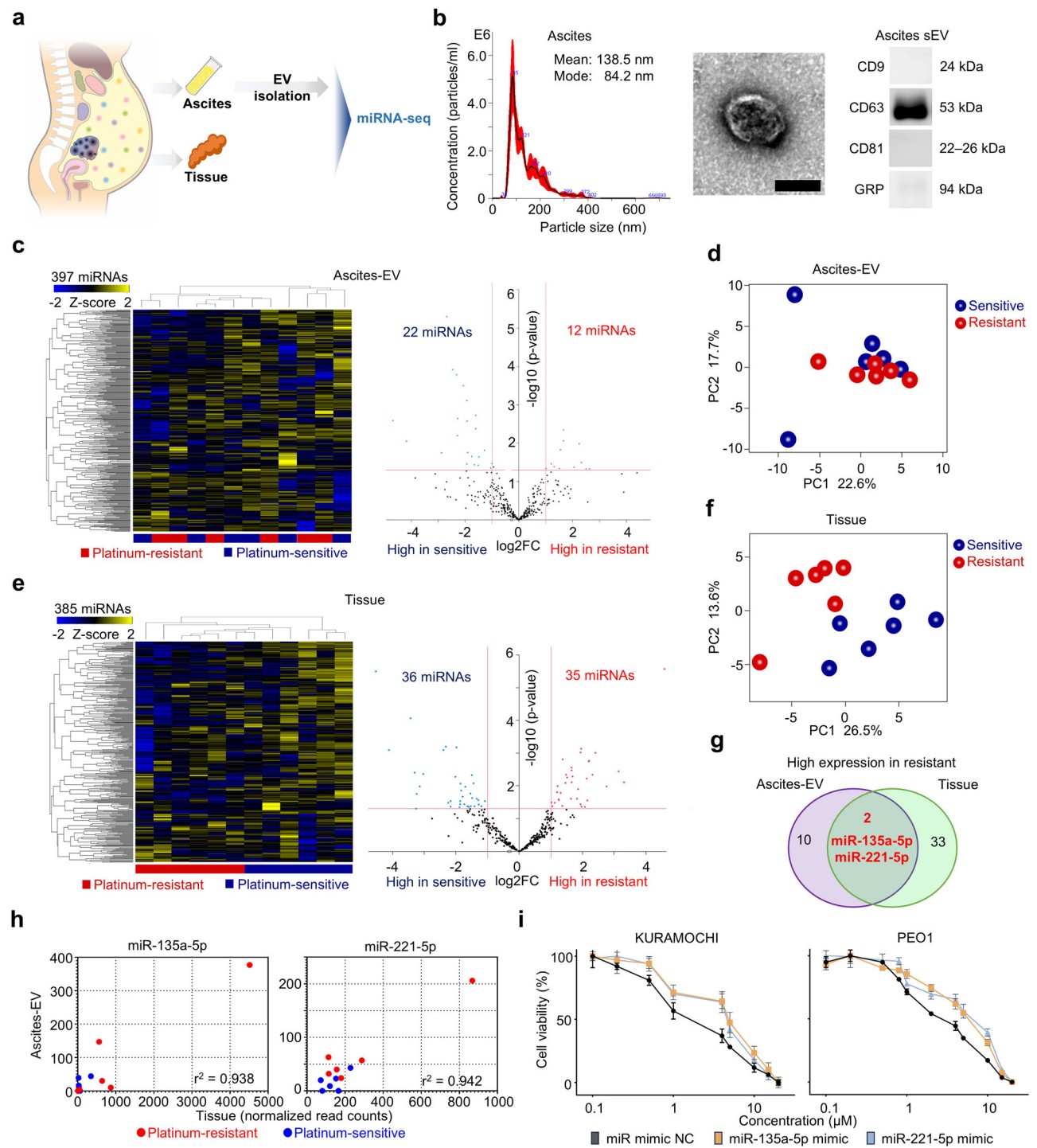

mesothelium, are immortalized with mutant CDK4, cyclin D1, and TERT using lentivirus-mediated gene transfer. The American Type Culture Collection supplied MeT-5A. The European Collection of Authenticated Cell Cultures provided HFFF2. MTK and Met5A cells were cultured in Roswell Park Memorial Institute (RPMI) 1640 (Nacalai Tesque) supplemented with 10% fetal bovine serum (FBS) and 1% penicillin–streptomycin (PS). HFFF2 cells were cultured in Dulbecco's Modified Eagle Medium (4500 mg/liter of glucose) (Nacalai Tesque) supplemented with 10% FBS and 1% PS. Cells were used during the five passages after primary culture in subsequent experiments. The cell lines tested negative for *Mycoplasma* contamination and were utilized between five and 40 passages for the experiments.

## Isolation of EVs from cell lines

The EV isolation method utilized in this study adhered to the standard indications of the International Society for EVs[39,40]. Cultured cells were washed twice in PBS 24 h after passage to prepare them for EV isolation. MTK cell was incubated in advanced RPMI 1640 medium (Thermo Fisher Scientific) containing 1% PS. A conditioned medium (CM) was collected after 48 h of incubation. CM was ultracentrifuged at 500 g for 15 min at 4 °C to remove cell debris, and the supernatant was collected and used for EV purification. The CM was filtered using a 0.22 μm filter and ultracentrifuged at 32,000 rpm for 120 min at 4 °C using an SW32 Ti rotor (Beckman Colter). The pellet was suspended in PBS and ultracentrifuged twice with the same conditions, and the sEV pellet was resuspended in PBS.

**Fig. 5 | Identifying unique EV-miRNA profiles of PROC. a** Study design to determine EV-miRNA profiles based on initial surgical tissue samples and ascites of HGSOC. **b** Characterization of the EVs from patients' ascites. Nanoparticle tracking analyses demonstrate the particle size of the EVs. Transmission electron microscopy was utilized to visualize the EVs. The scale bar indicates 100 nm. Immunoblot analysis for CD9, CD63, CD81, and GRP of representative EV samples from patient ascites. **c** Hierarchical clustering and heatmap showing 397 differentially expressed miRNA in ascites-EV between platinum-sensitive and platinum-resistant. The differentially expressed miRNA was defined as an absolute log2 fold change exceeding 1. Volcano plot demonstrating significantly differentially expressed miRNA in ascites-EV between platinum-resistant and platinum-sensitive, based on a cut-off of |log2FC| of >1 and an adjusted *P*-value of <0.05. Adjusted *P*-values for each miRNA were calculated using the Wald test in DESeq2. **d** Principal component analysis of miRNA sequencing in ascites-EV, including 6 platinum-resistant and 6 platinum-sensitive samples. **e** Hierarchical clustering and heatmap illustrating 385 differentially expressed miRNA in HGSOC tissues between platinum-sensitive and

resistant. The differentially expressed miRNA was defined as an absolute log2 fold change of >1. Volcano plot illustrating significantly differentially expressed miRNA in ascites-EV between platinum-resistant and platinum-sensitive, based on a cut-off of |log2FC| of >1 and an adjusted *P*-value of <0.05. Adjusted *P*-values for each miRNA were calculated using the Wald test in DESeq2. **f** Principal component analysis of miRNA sequencing in HGSOC tissues, including six platinum-resistant and six platinum-sensitive samples. **g** Venn diagram based on volcano plots (**c**) and (**e**) demonstrating miRNAs with high expression in both resistant ascites-EV and resistant tissues. **h** Correlation of miRNA expression in ovarian cancer tissues and ascites-EV based on miRNA sequencing data. R-values were calculated from Spearman's correlations. **i** Cisplatin sensitivity of transfected KURAMOCHI and PEO1 cells measured using the MTS assay. KURAMOCHI cells were transfected with a 20-nM miRNA mimic for 24 h, and subsequently, cells were treated with a cisplatin-containing medium for 48 h. PEO1 cells were transfected with a 20 nM miRNA mimic for 24 h, and subsequently, cells were treated with a cisplatin-containing medium for 72 h.

## Protein quantification and size analysis of EVs

A Qubit protein assay kit (Thermo Fisher Scientific) with the Qubit 4.0 Fluorometer (Invitrogen Co., MA, USA) was used to quantify the protein concentration of EVs and cell lysates, following the manufacturer's protocol. A NanoSight NS300 (Malvern Panalytical Ltd., UK) nanoparticle tracking analyzer was used to analyze the size distribution and particle concentration in the EV preparations. Samples were diluted in PBS and injected into the measuring chamber, and EV flow was recorded in triplicate measurements (30 s each) at room temperature. Equipment settings for data acquisition were kept constant between measurements, with the camera level set to 13. Characterization of EVs.

## Western blotting analysis

EV samples and cell lysate, prepared with adjusted amounts of protein, were loaded onto polyacrylamide gels for electrophoretic separation of proteins at 20 mA. After blocking with Blocking One (Nacalai Tesque Inc., Japan) and skim milk (Snow Brand Megmilk Co., Japan) for 1 h at room temperature, the membranes were incubated overnight at 4 °C with the following primary antibodies: mouse monoclonal anti-CD9 (CBL162, Merck; dilution 1:100), rabbit monoclonal anti-CD63 (EXOAB-CD63A-1, System Biosciences LLC, CA, USA; dilution 1:1000), mouse monoclonal anti-CD81 (sc-166029, Santa Cruz Biotechnology, TX, USA; dilution 1:100), and mouse monoclonal anti-GRP94 (sc-393402, Santa Cruz Biotechnology; dilution 1:100), mouse monoclonal anti-FOLR1 (MAB5646, R&D Systems; 5 μg/ml), rabbit monoclonal anti-CLDN3 (#83609S, Cell Signaling Technology; dilution 1:500), recombinant anti-TACSTD2 (#AB214488; Abcam; dilution 1:2000), anti-JAK1 rabbit monoclonal antibody (#3344, Cell Signaling Technology, 1:1000), phospho-Jak1 rabbit antibody (#3331, Cell Signaling Technology, 1:1000), anti-JAK2 rabbit antibody (F0231, Selleck, 1:1000), phospho-Jak2 rabbit antibody (#3776, Cell Signaling Technology, 1:1000), JAK3 rabbit antibody (#8827, Cell Signaling Technology, 1:1000), phospho-JAK3 rabbit antibody (#5031, Cell Signaling Technology, 1:1000), TYK2 rabbit antibody (F0864, Selleck, 1:1000), phospho-TYK2 rabbit antibody (#68790, Cell Signaling Technology, 1:1000), and anti β-Actin antibody (99361, FUJI-FILM Wako, 1:10,000). The membranes were washed for 5 min three times using tris-buffered saline with 0.1% Tween 20 and then incubated for 1–3 h at room temperature with secondary horseradish peroxidase-conjugated mouse anti-rabbit IgG (NA934-1ML, Cytiva Lifesciences, USA; dilution 1:5000) or anti-mouse IgG (NA931-1ML, Cytiva; dilution 1:2000) antibodies. ImageQuant LAS 4010 (GE Healthcare, IL, USA) was then used to image the membranes.

## Transmission electron microscopy

EV pellets were resuspended in PBS. A hydrophilic treatment device (PIB-10, Vacuum device, Mito, Japan) was used to hydrophilize a mesh grid with carbon support film (Nissin-EM, Tokyo, Japan), and the surface of the support membrane was attached to a 5-μL drop of the EV sample for 10 s. The sample was blocked with 1% bovine serum albumin for 1 h, washed five

times with PBS, fixed with 1% glutaraldehyde for 10 min, washed five times with ultrapure water, and stained with uranium for 10 s. The samples were dried thoroughly and visualized using a transmission electron microscope (JEM-1400 Plus, JEOL Ltd., Tokyo, Japan).

## EV treatment

KURAMOCHI cells were seeded in 6-well plates at a $5.0 \times 10^5$ cells/well (2 mL) density. The following day, approximately 25 μg of EV extracted from MTK cells/100 μL of PBS or 100 μL of PBS as a control were added. The cells were washed twice with PBS and collected in QIAzol after a 48 h exposure. RNA was extracted as previously described. Isolated EVs derived from MTK cells were quantified using the Qubit protein assay (Thermo Fisher Scientific) on the day before EV uptake observations, and 3 μg of EVs were added to PBS for a total of 200 μL. The same volume of PBS was used as a control. CellMask Green Plasma Membrane Stain (Thermo Fisher Scientific) was added to each tube at a 1:200 ratio, which was then heated for 30 min at 37 °C. The tubes that contain EVs were ultracentrifuged at 12,000 g using a TLA-55 rotor (Beckman Colter) for 1 h at 4 °C. The pellets were washed with PBS, centrifuged at 12,000 g for 1 h at 4 °C, and resuspended in 30 μL of PBS. KURAMOCHI cells were seeded in a μ-Slide 8 Well Glass Bottom (Ibidi, Gräfelfing, Germany) plate at $2 \times 10^4$ cells/well. During the observation day, 3 μg of stained EVs were added per well. The cells were washed with PBS after 3 h, and CellMask Deep Red Plasma Membrane Stain (Thermo Fisher Scientific) was added to the medium of each well at a 1:1000 ratio and incubated for 10 min at 37 °C. The cells were washed with PBS, and Cellstain Hoechst 33,342 solution (DOJINDO, Kumamoto, Japan) was added to the medium at a 1:2000 ratio, followed by incubation for 10 min at 37 °C. Finally, the cells were gently washed twice with PBS and visualized using confocal laser scanning microscopy (AX/AXR, Nikon, and Tokyo, Japan).

## RNase and Triton X-100 treatment

The collected EVs were divided into three equal volumes: a negative control; RNase only; and Triton X-100 and RNase. EVs were treated with or without 0.1% Triton X-100 (Thermo Fisher Scientific) for 30 min at room temperature, followed by treatment with RNase A (5 μg/mL, Nippon gene) for 15 min at 37 °C. After RNase and/or Triton X-100 treatment, RNA extraction and quantitative reverse transcription polymerase chain reaction were performed.

## Chemicals

Selleck (Peficitinib and Ruxolitinib) provided all selective JAK. Additionally, cisplatin (Nichi-Iko Pharmaceuticals) was used. Excluding cisplatin, the drugs were dissolved in DMSO as stock solutions and further diluted in the culture medium for experiments.

## siRNAs

Silencer Select Pre-designed siRNAs for each gene and Negative Control No.2 siRNA (Thermo Fisher Scientific) were used. The assay IDs were

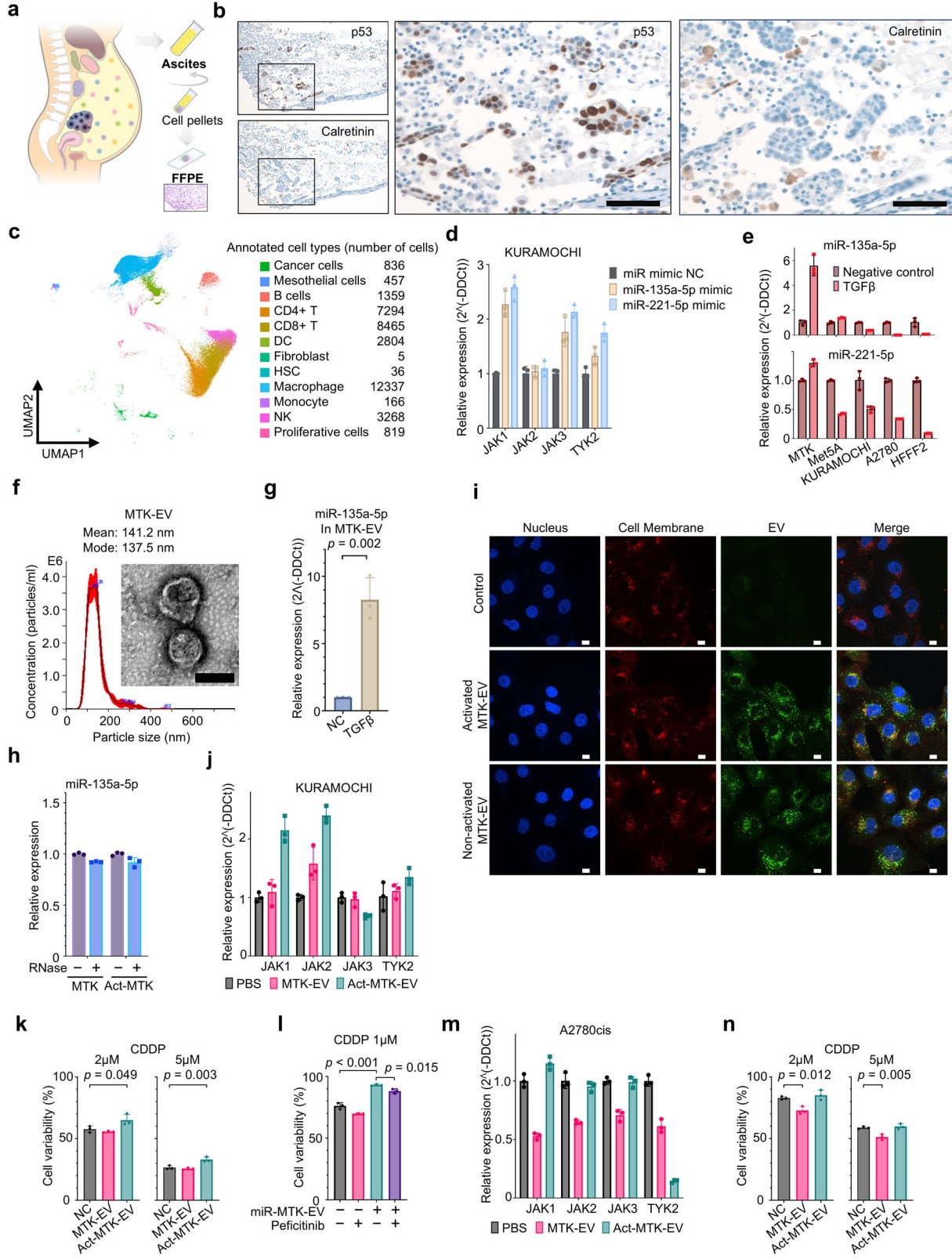

s7646 (siJAK1 No. 1), s7647 (siJAK1 No. 2), s7651 (siJAK2 No. 1), s7649 (siJAK2 No. 2), s532723 (siJAK3 No. 1), s532724 (siJAK3 No. 2), s14537 (siTYK2 No. 1), and s14535 (siTYK2 No. 2). Cells were transfected with 3 nmol/L of siRNA using Lipofectamine RNAi Max (Thermo Fisher Scientific).

## miRNA mimics

In the present study, mirVana miRNA mimics (Thermo Fisher Scientific) were used to induce miRNA overexpression. The assay IDs were miR-135a-5p (MC11126), miR-221-5p (MC12613), and Negative Control (4464058). Cells were transfected with 20 nM of miRNA mimics using

**Fig. 6 | Mesothelial cells derived EVs regulated platinum sensitivity in ovarian cancer. a** Study design for analysis of cells in ascites. FFPE blocks from cell pellets were obtained after centrifuging HGSOC ascites. **b** Representative images of immunohistochemical staining for p53 and calretinin of FFPE prepared as in (**a**). Scale bars indicate 100 μm. **c** UMAP plots showing ascites (*n* = 8) from patients with HGSOC from the single-cell RNA sequencing dataset (PRJCA005422). **d** Relative expression of the *JAK* family mRNA in KURAMOCHI after transfection of miR-135a-5p and miR-221-5p. **e** Relative expression of miR-135a-5p and miR-221-5p in each cell after TGFβ exposure. **f** EV characterization from MTK cell culture medium. Nanoparticle tracking analyses demonstrating the particle size of the EVs. Transmission electron microscopy was utilized to visualize the EVs. The scale bar indicates 100 nm. **g** Relative expression of miR-135a-5p in EV-derived MTK cell culture medium after TGFβ exposure for 12 h. NC: negative control. **h** Relative expression of

miR-135a-5p in EVs treated with/ without RNaseA. **i** Representative images of KURAMOCHI cells taking up EVs derived from activated or non-activated MTK cell culture medium, visualized using confocal laser scanning microscopy. The scale bar indicates 10 μm. **j** Relative JAK family mRNA expression in KURAMOCHI after MTK-EV and activated MTK-EV treatment for 24 h. **k** Cell viability after cisplatin treatment of KURAMOCHI treated with MTK-EV or activated MTK-EV for 48 h, followed by cisplatin for 72 h, as (Supplementary Fig. 6i) protocol. **l** Cell viability after cisplatin treatment of KURAMOCHI treated with/without miR-enriched MTK-EVs (miR-MTK-EV) for 24 h, followed by cisplatin with/without Peficitinib for 48 h. **m** Relative *JAK* family mRNA expression in A2780cis after MTK-EV and activated MTK-EV treatment for 24 h. **n** Cell viability after cisplatin treatment of A2780cis treated with MTK-EV or activated MTK-EV for 24 h, followed by cisplatin for 48 h, as (Supplementary Fig. 8a) protocol.

Lipofectamine® RNAi Max (Thermo Fisher Scientific) at 37 °C for at least 24 h.

## Cell viability assay
Cells were seeded into 96-well plates. After attachment, the cells were immediately treated with the inhibitors and incubated for 72 h. Cell viability was evaluated using the CellTiter-Glo 2.0 Cell Viability Assay (Promega). Luminescence measurements were taken 10 min after adding the reagent with a microplate reader (Molecular Devices). Viability was calculated with the percentage of untreated cells, and experiments were conducted in triplicate. The IC50 value was calculated using the following equation: $IC50 = 10[\log(A/B) \times (50 - C)/(D - C) + \log(B)]$, where A and B represent the highest and the lowest concentrations to cover an estimated IC50 value, respectively, and where C and D denote the cell viability at concentrations B and D, respectively.

## Cell proliferation assay
A2780cis and PEO1-CDDP cells were seeded in 96-well plates at a 2000 cells/well density. SiRNAs were transfected 24 h after seeding, and cells were treated with cisplatin or inhibitors 48 h after transfection. Cell confluence was measured every 6 h until 3 days using a live cell imaging system, IncuCyte SX5 (Sartorius, Johnson Avenue, Bohemia, USA).

## Quantitative reverse transcription polymerase chain reaction (qRT-PCR)
A ReverTra Ace qPCR RT Kit (Toyobo, Osaka, Japan) and TB Green Premix Ex Taq (Takara Bio, Shiga, Japan) were used for mRNA, following the manufacturer's instructions. Afterward, qPCR was performed using Mx3000P (Agilent Technologies), and the amplification program was denaturation at 95 °C for 30 s, followed by 40 amplification cycles of 95 °C for 5 s and 60 °C for 30 s. The amplified product was monitored by SYBR Green I dye fluorescence intensity, and GAPDH or β-actin was utilized as a reference gene to normalize the expression. The primer sequences were described in Supplementary Table 2. The TaqMan Advanced miRNA cDNA Synthesis Kit (Thermo Fisher Scientific) was used to synthesize cDNA for miRNAs, following the manufacturer's protocol. Quantitative PCR was conducted using the TaqMan Fast Advanced Master Mix and TaqMan Advanced miRNA Assay (Thermo Fisher Scientific). The assay IDs were miR-135a-5p (478778_mir) and miR-221-5p (478581_mir). The amplification program was denaturation at 95 °C for 10 min, followed by 40 amplification cycles at 95 °C for 15 s and 60 °C for 60 s. The amplified products were monitored by measuring FAM fluorescence intensity.

## Immunohistochemical staining
A microtome was utilized to prepare 4 μm-thick sections. The sections were deparaffinized and rehydrated, and antigen retrieval was achieved by incubating in 10 mM of citrate buffer (pH 6.0) at 95 °C for 25 min. Endogenous peroxidase activity was blocked with $H_2O_2$ in methanol for 20 min at room temperature. The sections were blocked with 10% rabbit, mouse, or goat serum and incubated overnight at 4 °C with primary antibody. Primary

antibodies included anti-JAK1 rabbit monoclonal antibody (#3344, Cell Signaling Technology, 1:100), anti-STAT1 rabbit monoclonal antibody (#14994, Cell Signaling Technology, 1:2000), anti-STAT3 mouse monoclonal antibody (#9139, Cell Signaling Technology, 1:500), anti-p53 mouse monoclonal antibody (sc126, Santa Cruz, 1:500), anti-Calretinin antibody rabbit polyclonal antibody (ab702, abcam, 1:100), Cleaved Caspase-3 monoclonal rabbit antibody (#9664, Cell Signaling Technology, 1:100), and anti-RIP3 polyclonal rabbit antibody (ab56164, abcam, 1:100). SAB-PO kits (Nichirei Biosciences, Tokyo, Japan) were used to incubate the sections with secondary antibody and peroxidase-labeled streptavidin for 30 min at room temperature. The sections were developed using a DAB solution (Nichirei Biosciences). They were incubated with hematoxylin, dehydrated, and mounted after rinsing with water. A ZEISS Axio Imager (Carl Zeiss, Oberkochen, Germany) was used to photograph the slides to create digital images, which were quantified using QuPath.

## Animal studies
The Nagoya University Institutional Animal Experimentation Committee (Approval No. 230137) reviewed and approved all animal procedures, which complied with the ARRIVE guidelines and were constructed following the UK Animals (Scientific Procedures) Act, 1986, and related guidelines. We have complied with all relevant ethical regulations for animal use. The experiments utilized 7-week-old female BALB/c nude mice. Luciferase-labeled A2780 and A2780cis cells were transplanted, and $5 \times 10^6$ cells/50 μL were injected intraperitoneally. Mice were treated intraperitoneally with 3.0 mg of peficitinib every 2 days for a total of five treatments. The mice were administered D-luciferin (Promega, Madison, WI) by intraperitoneal injection for in vivo imaging. After 10 min, photons in the whole bodies of the animals were measured by evaluating bioluminescence with an IVIS Spectrum imaging system (Caliper Life Science, Hopkinton, MA). LIVINGIMAGE 4.4 software (Caliper Life Science) was used to analyze the data. We developed PDX mouse models using tissues from HGS-PDSR and HGS-PDXS cases. Fresh surgical tissue was sectioned into ~3 mm three pieces and implanted subcutaneously into a 6-week-old female NOD.Cg-PrkdcscidIl2rgtm1Wjl/SzJ mouse (Charles River Laboratories Japan, Kanagawa, Japan). The generation that harbored the patient-derived material was termed F1, with subsequent generations numbered consecutively (F2, F3, F4, etc.). We used 7-week-old female Balb/c-nu/nu mice (Charles River Laboratories). We reimplanted cancer tissues of HGS-PDXR-F3 into mice subcutaneously under isoflurane anesthesia. Intraperitoneal administration of peficitinib was initiated when the long diameter of the subcutaneous tumor exceeded 5 mm. The modified ellipsoid formula (length × width2 × 0.5) was used to calculate tumor volume.

## Statistics and reproducibility
Statistical analysis was performed with GraphPad Prism ver. 10.2.0 for Windows, RStudio, and R software (ver. 4.4.0). The Welch's t-test was used to determine the significant difference between the means of two sets of data. The paired *t* test was used to determine the significant difference between paired two sets of data. Dunnett's test was used for multiple comparisons

with the control group using the multicomp package (ver. 1.4-17). To evaluate gene expression correlation, the Pearson correlation coefficient was calculated. The Kaplan–Meier curves and log-rank test were used for the survival analysis. A P value of less than 0.05 was considered statistically significant.

## Data availability

The data generated in this study are available in Gene Expression Omnibus (GEO) at Super Series GSE274659. All other data are available from the corresponding author on reasonable request.

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

## Acknowledgements

We express our gratitude to the members of the Department of Obstetrics and Gynecology at Nagoya University Graduate School of Medicine. We received technical assistance from Division of Medical Research Engineering at Nagoya University Graduate School of Medicine. This work was financially supported by a Grant-in-Aid for Scientific Research, the Japan Society for the Promotion of Science (JSPS KAKENHI Grant Numbers 21H03075, 22K18394 and 24K02586), Nozawa Memorial Research Grant,

the Fusion Oriented Research for Disruptive Science and Technology (FOREST; JPMJFR204J) from the Japan Science and Technology Agency, and the Tokai Pathways to Global Excellence (T-GEx), part of the MEXT Strategic Professional Development Program for Young Researchers.

## Author contributions

.A.Y. and K.S. designed the study. K.S., A.Y., K.Y., M.K., E.I., and S.M. performed experiments, data acquisition and interpretation. K.S., H.S., M.Y., S.T., N.Y., and K.N. collected the data. K.S., T.S., and S.Y. provided all the clinical samples, data analysis and pathological interpretation. K.S., K.Y., Y.Y., and H.S. performed bioinformatic analyses. A.Y. and Y.Y. designed the experiments, analyzed the data, conducted scientific direction. A.Y. and H.K. provided supervision. K.S. wrote the original draft and all authors reviewed and edited the paper. All authors read and agreed to the published version of the article.

## Competing interests

The authors declare no competing interests.
