## [Transparent Peer Review file · Communications Biology]

Overcoming platinum-resistant ovarian cancer targeting the activated JAK-STAT pathways via extracellular vesicles

Corresponding Author: Dr Akira Yokoi

This manuscript has been previously submitted at another journal. This document only contains information relating to versions considered at Communications Biology.

Version 0:

Reviewer comments:

Reviewer #1

(Remarks to the Author)

The authors revealed that the clinical samples from chemo-naive patients with HGSOC contained mesothelial cell-derived miR-135a-5p-enriched extracellular vesicles (EVs) in ascites and these EVs activate the Janus kinase/signal transducer and activator of transcription (JAK-STAT) pathway, causing platinum resistance in cancer cells. Furthermore, the authors showed that JAK inhibitors have antitumor effects for platinum resistance and synergistic effects with platinum. This study showed that EVs in ascites regulated platinum sensitivity in ovarian cancer cells, and JAK targeting therapeutic strategy could overcome PROC. The experimental design of this study is well organized and the results presented novel strategy for drug resistant ovarian cancer. Please clarify some minor points as listed below:

1. In legend of Fig.1j, please describe what is HGS-76R, HGS-36R, HGS-72S, and HGS-36S. How many tissue samples in total are you tested in this experiment on platinum-sensitivity and resistance. Two of each?
2. In the in vivo experiment of JAK inhibitor as shown Fig.4, although the number of animals is limited (N=3), the results look fine. The question is, is the tumor suppression effect in this animal due to cell death of the cancer cells, and if so, what is the evidence for necrosis or apoptosis?
3. Upon the EVs from the patients' ascites/tissue, are they have any ovarian cancer-specific markers? Profile of the conventional CD markers are different from CAOV3 to Ascites EVs.
- 4.

Reviewer #2

(Remarks to the Author)

This study demonstrates that activated mesothelial cells enriched with miR-135a-5p-EV targeting JAK/STAT pathway to induce platinum resistance in ovarian cancer, which is interesting and significant. However, there are some issues that need to be addressed.

1. Many figures lack statistical information, such as fig1f, 1i, fig3, etc. The authors should add these and provide clear explanations in the figure legends.
2. The results for STAT family in tissues and A2780 cell lines appear to be contradictory (see fig2 and extended data fig3a), which is not mentioned. The authors should explain it.
3. Fig3a and 3b: The protein level of the JAK family should be tested.
4. JAK proteins typically function through phosphorylation. Has phosphorylation been detected in tissues or cells?
5. Lines 173-175: The survival analysis of miRNA shows no statistical significance.
6. Lines 179-180: Is the expression "in the cancer cell lines in other ovarian cancer cell lines" correct? Similarly, there is no statistical information.
7. There are color errors in the icons of Fig6d, extended data fig6, and 6c, which is confusing.
8. The results for mesothelial cells MTK and Met-5A are inconsistent in fig6e. Can the authors explain it? Could it be because MET-5A originates from the pleural membrane?
9. Fig6f and extended data fig6f: TEM, particle size analysis, and Western blotting for activated MTK-EVs are needed. It states "representative samples of ascites EVs" in the figure legend for extended data fig6f, but there are no ascites EVs in

extended data fig6f.

10. A non-activated MTK-EV control is needed for fig6i.

11. Experiments involving the co-transfection of miR-135a-5p-enriched EVs with JAK1, 2, 3, TYK2 knockdown or peficitinib should be added.

12. Many results lack consistency, such as fig3a TYK2, fig6d JAK2, fig6j JAK3, fig6o TYK2, which is overall doesn't make the present data too convincing. The author should explain it.

13. A non-EV control is needed when ovarian cancer cell lines are treated with MTK-EV or activated MTK-EV.

14. Lines 217-219 are confusing and should be rephrased.

15. Line 451: It seems I haven't seen any results for miRNA inhibitors.

16. Line 482: The reference gene for miRNA in cells or EVs should be stated.

17. The schematic diagram should distinguish between non-activated and activated mesothelial cells for easy understanding.

Reviewer #3

(Remarks to the Author)

In this study, the authors establish a correlation between the PROC and the JAK-STAT pathway using bulk RNA sequencing. These results (i.e., upregulation of the JAK-STAT family) were then independently confirmed in patient-derived samples through immunohistochemistry. The authors then tested the effect of JAK pathway inhibition *ex vivo* and *in vivo* in well-established mouse models using a known drug. Remarkably, the research group isolated extracellular vesicles (EVs) emanating from ascites and cancer tissues. Through sequencing, they identified two particular miRNAs that were enriched in the EVs from the resistant groups. They independently demonstrated that transfection of these two miRNAs (independent of EV transfer) contributed to cisplatin resistance. Using EVs enriched for these miRNA species, they showed that EV exposure to the cancer cell model decreased sensitivity to cisplatin.

The effect of cisplatin sensitivity related to the two proposed miRNAs is consistent with the known roles of related miRNAs (reviewed for instance in Yingchun Shao, Shuangshuang Zhang, Yuxin Pan, Zhan Peng, Yinying Dong, miR-135b: A key role in cancer biology and therapeutic targets, *Non-coding RNA Research*, Volume 12, 2025). Importantly, the study integrates *ex vivo* and *in vivo* validation and establishes a link between tumor-derived EVs and platinum resistance, adding to the growing body of literature on the role of EVs in oncology. The study appears technically sound to this reviewer.

The only minor concern is the need to formally establish whether the two enriched miRNAs are truly encapsulated within the EVs or associated with the surface. It was not sufficiently clear in the methods section whether this possibility was tested.

Establishing formally the EV-mediated delivery would be ideal but remain challenging. As this later point remains controversial, perhaps this should be clearly added in the paragraph already highlighting the limitation of the study.

Version 1:

Reviewer comments:

Reviewer #1

(Remarks to the Author)

The authors revised their manuscript as according to the comments.

Reviewer #2

(Remarks to the Author)

The authors addressed my comments well. I have no other questions. I suggest accept it for publication .

Reviewer #3

(Remarks to the Author)

The authors have satisfactorily addressed my previous concerns. The manuscript is significantly improved after the revisions made in response to the other reviewers' comments

POINT-BY-POINT RESPONSE TO THE REVIEWERS.

(Our responses are in *italics and blue*, and the line numbers refer to the document after changes.)

Reviewers' comments:

Reviewer #1 (Remarks to the Author):

The authors revealed that the clinical samples from chemo-naive patients with HGSOC contained
mesothelial cell-derived miR-135a-5p-enriched extracellular vesicles (EVs) in ascites and these EVs
activate the Janus kinase/signal transducer and activator of transcription (JAK-STAT) pathway,
causing platinum resistance in cancer cells. Furthermore, the authors showed that JAK inhibitors
have antitumor effects for platinum resistance and synergistic effects with platinum. This study
showed that EVs in ascites regulated platinum sensitivity in ovarian cancer cells, and JAK targeting
therapeutic strategy could overcome PROC. The experimental design of this study is well organized
and the results presented novel strategy for drug resistant ovarian cancer. Please clarify some minor
points as listed below:

1. In legend of Fig.1j, please describe what is HGS-76R, HGS-36R, HGS-72S, and HGS-36S. How
many tissue samples in total are you tested in this experiment on platinum-sensitive and resistance.
How many of each?

*Thank you for your comment. In Visium spatial transcriptomics, a total of four tissue samples were*
*tested, including two samples from platinum-resistant cases and two from platinum-sensitive cases.*
*We have revised the legend of Figure 1j.*

*j) Top: H&E staining of high-grade serous ovarian cancer (HGSOC) tissue sections from platinum-*
*resistant cases (HGS-76R and HGS-36R) and platinum-sensitive cases (HGS-72S and HGS-36S).*
*Supplementary Table 1 shows the clinical information of the patients.*

2. In the in vivo experiment of JAK inhibitor as shown Fig.4, although the number of animals is
limited (N=3), the results look fine. The question is, is the tumor suppression effect in this animal
due to cell death of the cancer cells, and if so, what is the evidence for necrosis or apoptosis?

*Thank you for highlighting this important aspect. To verify whether necrosis or apoptosis was*
*involved, immunohistochemistry for cleaved caspase-3 and RIPK3 was performed on tumors from in*
*vivo experiments. The detection of cleaved caspase-3 positivity alongside RIPK3 negativity suggests*
*that the JAK inhibitor induced apoptosis, a mechanism consistent with findings from previous studies*
*(Xiong H, et al. Neoplasia. 2008., Li X, et al. Oncol Lett. 2021.). These results have been included in*
*the Results section.*

*Lines 156–159: “The anticancer effect was supposed to be the result of apoptosis induced by*
*Peficitinib, according to immunostaining of the excised tumor, which was positive for cleaved*
*caspase-3 and negative for RIPK3 (Supplementary Fig. S4d).”*

*Supplementary Figure. S4d Legend: Representative images of immunohistochemistry of cleaved caspase-3 and RIPK3 in*
*the excised tumors (from Fig4. a, d and Supplementary Figure. 4a). Scale bars indicate 100 μm.*

3. Upon the EVs form the patients' ascites/tissue, are they have any ovarian cancer-specific markers?
Profile of the conventional CD markers are differ from CAOV3 to Ascites EVs.

*As you pointed out, the conventional CD marker profile of EVs derived from ascites samples was*
*distinct from that of CAOV3 cell line-derived EVs. As demonstrated in our previous study (Yokoi A, et*
*al. Sci Adv. 2023), we identified FRα, Claudin-3, and TACSTD2 as high-grade serous ovarian cancer*
*(HGSOC)-specific sEV proteins. Notably, the profiles of these markers were consistent between EVs*
*derived from the CAOV3 cell line and EVs derived from ascites, as shown in the newly added*
*Supplementary Figure 5a.*

*Lines 165–167: “Ascites-EVs were isolated following the protocol and characterized by nanoparticle*
*tracking analysis, transmission electron microscopy, and western blotting (Fig. 5b and*
*Supplementary Fig. 5a).”*

*Figure. 5b* *Supplementary Figure. S5a*

*Figure. 5b Legend: Immunoblot analysis for CD9, CD63, CD81, and GRP of representative sEV samples from*
*patient ascites.*

*Supplementary Figure. S5a Legend: Characterization of EVs from patient ascites. Immunoblot analysis for FRα,*
*Claudin-3, and TACSTD2 of representative sEV samples.*

Reviewer #2 (Remarks to the Author):

This study demonstrates that activated mesothelial cells enriched with miR-135a-5p-EV targeting
JAK/STAT pathway to induce platinum resistance in ovarian cancer, which is interesting and
significant. However, there are some issues that need to be addressed.

1. Many figures lack statistical information, such as fig1f, 1i, fig3, etc. The authors should add these
 and provide clear explanations in the figure legends.

*Thank you for the reviewer's valuable suggestion regarding the inclusion of statistical information in
 the figures. We have revised the Statistical Analysis section. The results of the statistical analysis are
 presented in Figure 1f, 1i, 3a, 4e, and other figures.*

*Lines 555–561: “Statistical analysis was performed with GraphPad Prism ver. 9.3.1 for Windows,
 RStudio, and R software (ver. 4.2.2). The Welch's t-test was used to determine the significant
 difference between the means of two sets of data. The paired t test was used to determine the
 significant difference between paired two sets of data. Dunnett's test was used for multiple
 comparisons with the control group using the multcomp package (ver. 1.4-17). To evaluate gene
 expression correlation, the Pearson correlation coefficient was calculated. The Kaplan–Meier curves
 and log-rank test were used for the survival analysis. A P value of less than 0.05 was considered
 statistically significant.”*

*Figure 1f*

Figure 1i

*Figure 3a*

Figure 4e

2. The results for STAT family in tissues and A2780 cell lines appear to be contradictory (see fig2
 and extended data fig3a), which is not mentioned. The authors should explain it.

*Thank you for highlighting this point. As you observed, there appear to be conflicting trends in the
 expression and activation of the STAT family between tissue samples (Figure 1f) and the A2780 cell
 line in our data (Figure 2a and Supplementary Figure 3a). These differences are likely due to the
 distinct biological contexts of tumor tissues versus cell line models. Clinical samples contain not*

only cancer cells but also various stromal and immune cells, with complex signaling mediated by
EVs. In contrast, the A2780 cell line represents a monoclonal-homogeneous cancer cell population,
lacking the influence of the tumor microenvironment, which may account for the observed differences
in STAT family activation. To strengthen the complementarity and translational relevance of our
findings, we subsequently performed experiments using both in vitro and in vivo models
characterized by high expression of the JAK/STAT family. Specifically, we performed in vitro
experiments using the platinum-resistant ovarian cancer cell line PEO1-CDDP, which exhibits high
expression of STAT family members. In parallel, we performed in vivo experiments using a patient-
derived xenograft model with high expression of the JAK/STAT family. These approaches allowed us
to validate the functional significance of JAK/STAT signaling in platinum resistance in different
experimental systems, thereby strengthening the robustness of our conclusions. We have revised the
Discussion section accordingly.

Line 286–290: “Second, in our in vitro experiments, not all JAK family members exhibited similar
expression changes. This may be due to the limitations of cell lines, which reflect uniform,
monoclonal changes. Additionally, JAK-STAT signaling can be activated through JAK
phosphorylation as mono- or heterodimers, even if some members are not highly expressed.”

3. Fig3a and 3b: The protein level of the JAK family should be tested.

We appreciate the reviewer's important suggestion regarding the evaluation of JAK family proteins at
the protein level. In response, we conducted Western blot analysis in both siRNA-treated and
untreated ovarian cancer cell lines. These experiments confirmed the reduction of target JAK
proteins at the protein level (Supplementary Figs. S3b and S3c). For JAK3, although multiple
antibodies were tested, consistent and clear results could not be obtained. Therefore, we have
included the most reliable result obtained for JAK3 among those tested in the revised manuscript.

Supplementary Fig. S3c legend: Evaluation of transfection efficiency of two siRNAs targeting the JAK family (siRNA No.
1 and No. 2). Western blot analysis of whole-cell lysates from A2780cis and PEO1-CDDP cells, probed for JAK1, JAK2,
JAK3, TYK2, and β-actin.

4. JAK proteins typically function through phosphorylation. Has phosphorylation been detected in
tissues or cells?

Thank you for this valuable observation. To address this point, we conducted western blot analysis to
assess the phosphorylation status of JAK proteins in ovarian cancer cell lines. At the protein level,
we observed phosphorylation of JAK2 and JAK3 in A2780cis cells, whereas phosphorylation of
JAK1 was confirmed in PEO1-CDDP cells. These findings are described in lines 134–138 of the
revised manuscript, and the original blot images are included in Supplementary Figure S3b.

*Lines 134–138: “We examined the expression levels of JAK-STAT in paired ovarian cancer cell lines,*
*including parental and platinum-resistant lines. Several JAKs were found to be highly expressed at*
*the mRNA level in resistant cell lines (Fig. 3a). At the protein level, high expression of JAK1 as well*
*as phosphorylation of JAK2 and JAK3 were observed in A2780cis cells, while phosphorylation of*
*JAK1 was confirmed in PEO1-CDDP cells. (Supplementary Fig. S3b).”*

*Supplementary Fig. S3b legend: Immunoblotting analysis of whole-cell lysates from A2780, A2780cis, PEO1, and PEO1-*
*CDDP cells, probed for JAK1, pJAK1, JAK2, pJAK2, JAK3, pJAK3, TYK2, pTYK2, and beta-actin.*

5. Lines 173-175: The survival analysis of miRNA shows no statistical significance.

*Thank you for bringing this to our attention. The data in Supplementary Figure 5c represents the*
*analysis of miRNA expression in tumor tissues, which does not directly reflect the expression of*
*miRNAs within extracellular vesicles (EVs) in ascites, which is the primary focus of our study. We*
*have revised the manuscript accordingly.*

*Lines 177–180: “Although there was no statistically significant difference, high expression of both*
*miRNA types in ovarian cancer tissues tended to be associated with shorter overall survival based on*
*Kaplan–Meier curves (Supplementary Figure 5c).”*

6. Lines 179-180: Is the expression "in the cancer cell lines in other ovarian cancer cell lines"
correct? Similarly, there is no statistical information.

*Thank you for pointing out this error in phrasing and lack of statistical detail. The correct*
*explanation is as follows. The intended explanation has been clarified, and we have updated the*
*statistical analysis of IC50, including Supplementary Figure 5d in the revised manuscript.*

*Line 183–185: “Additionally, the IC50 of cisplatin increased by the transfection of miR-135a-5p and*
*miR-221-5p in other ovarian cancer cell lines (Fig. 5i and Supplementary Fig. 5d, e).”*

*Supplementary Fig. S5e legend: Cisplatin IC 50 values of KURAMOCHI, PEO1, and SKOV3 cells transfected with*
*miRNA mimics, measured using the MTS assay.*

7. There are color errors in the icons of Fig6d, extended data fig6, and 6c, which is confusing.

*We apologize for the confusion caused by the color inconsistencies in Figures 6d, Supplementary*
*Figure 6, and 6c. We have made the necessary corrections as suggested.*

*Fig.6d*

158 *Supplementary Fig.6a*

158 *Supplementary Fig.6c*

8. The results for mesothelial cells MTK and Met-5A are inconsistent in fig6e. Can the authors
explain it? Could it be because MET-5A originates from the pleural membrane?

*As you correctly noted, we suppose that the difference in the response to TGFβ is indeed due to the*
*distinct origins of the mesothelial cell lines: MTK is derived from peritoneal mesothelial cells, while*
*Met-5A originates from pleural mesothelial cells.*

9. Fig6f and extended data fig6f: TEM, particle size analysis, and Western blotting for activated
MTK-EVs are needed. It states "representative samples of ascites EVs" in the figure legend for
extended data fig6f, but there are no ascites EVs in extended data fig6f.

*Thank you for bringing this error to our attention. We have now included TEM, particle size*
*analysis, and Western blotting for activated MTK-EVs. These experiments were performed on sEVs*
*derived from cell lines, not ascites. We have corrected the figure legends accordingly.*

*Supplementary Fig. S6f, g legend: f) EV characterization from activated MTK cell culture medium. Nanoparticle*
*tracking analyses demonstrating the particle size of the EVs. Transmission electron microscopy was utilized to*
*visualize the EVs. The scale bar indicates 100 nm. g) Immunoblot analysis for CD9, CD63, CD81, and GRP of cell*
*lysate and sEV representative samples.*

10. A non-activated MTK-EV control is needed for fig6i.

*Thank you for this valuable suggestion. We have now included a non-activated MTK-EV control in*
*Figure 6i.*

*Fig. 6i legend: Representative images of KURAMOCHI cells taking up EVs derived from activated or non-activated MTK*
*cell culture medium, visualized using confocal laser scanning microscopy.*

11. Experiments involving the co-transfection of miR-135a-5p-enriched EVs with JAK1, 2, 3, TYK2
knockdown or peficitinib should be added.

*Thank you for highlighting this important aspect. In addition to Figure 6L, we performed*
*experiments in which EVs enriched for miR-135a-5p were co-treated with Peficitinib.*

*Figure 6l legend: Cell viability after cisplatin treatment of KURAMOCHI treated with/without miR-enriched MTK-EVs*
*for 24 h, followed by cisplatin with/without Peficitinib for 48 h.*

12. Many results lack consistency, such as fig3a TYK2, fig6d JAK2, fig6j JAK3, fig6o TYK2, which
is overall doesn't make the present data too convincing. The author should explain it.

*Thank you for pointing out the apparent inconsistencies in the expression or phosphorylation*
*patterns of individual JAK family members across different figures (e.g., Fig. 3a TYK2, Fig. 6d*
*JAK2, Fig. 6j JAK3, Fig. 6o TYK2). We acknowledge that not all four members of the JAK family*
*showed similar expression changes in our in vitro experiments. One reason for this variability is the*
*inherent limitations of cell line experiments, which often reflect uniform, monoclonal changes.*
*Additionally, the JAK-STAT pathway can be activated through various mechanisms, including mutual*
*phosphorylation of neighboring JAKs or by phosphorylation as mono- or heterodimers, even when*
*all JAK family members are not highly expressed (O'Shea JJ, et al. N Engl J Med. 2013., Babon JJ,*

*Let la. Biochem J. 2014., Garrido-Trigo A, et al. J Crohns Colitis. 2020.). We have revised the*
*Discussion section to elaborate on this point.*

*Line 286–290: “Second, in our in vitro experiments, not all JAK family members exhibited similar*
*expression changes. This may be due to the limitations of cell lines, which reflect uniform,*
*monoclonal changes. Additionally, JAK-STAT signaling can be activated through JAK*
*phosphorylation as mono- or heterodimers, even if some members are not highly expressed.”*

13. A non-EV control is needed when ovarian cancer cell lines are treated with MTK-EV or activated
MTK-EV.

*Thank you for pointing out this important control requirement. In response, we have included non-EV*
*controls in the experiments involving MTK-EV and activated MTK-EV treatments, as shown in*
*Figures 6j, 6k, 6m, 6n, and Supplementary Figures 6j and 8c. These controls help to clarify the EV-*
*specific effects and improve the experimental rigor and interpretation of our results.*

*Figure 6j*

Figure 6k

Figure 6m

Figure 6n

*Supplementary Figure 6j*

220 *Supplementary Figure 8c*

14. Lines 217-219 are confusing and should be rephrased.

*We apologize for the confusion. We have revised the Discussion section as follows.*

*Line 227–228: “This study revealed that the expression of miR-135a-5p in both ascites-EVs and*
*cancer tissues was upregulated in platinum resistance, indicating activation of the JAK-STAT*
*pathway in PROC.”*

15. Line 451: It seems I haven't seen any results for miRNA inhibitors.

*We thank the reviewer for the careful review and accurate observation. The reviewer's comment is*
*correct; as you pointed out, we did not use miRNA inhibitors in our experiments. Instead, we focused*
*on inducing miRNA overexpression using mimics, and the relevant methodological details have been*
*clarified in the manuscript.*

*Line475–479: “In the present study, mirVana miRNA mimics (Thermo Fisher Scientific) were used to*
*induce miRNA overexpression. The assay IDs were miR-135a-5p (MC11126), miR-221-5p*
*(MC12613), and Negative Control (4464058). Cells were transfected with 20 nM of miRNA mimics*
*using Lipofectamine® RNAi Max (Thermo Fisher Scientific) at 37°C for at least 24 h”*

16. Line 482: The reference gene for miRNA in cells or EVs should be stated.

*We acknowledge that identifying a single, universal reference gene for miRNA normalization in all*
*cell types or EVs remains a challenge. While RNU6 is sometimes used, its stability can vary*
*significantly depending on the sample type; for instance, RNU6 expression is not constant in ovarian*
*cancer tissue (Bignotti E, et al. J Cell Mol Med. 2016.). Given the absence of an absolute consensus*
*on EV-miRNA internal reference controls, we performed our experiments and analyses in accordance*
*with the MISEV2023 recommendations (J Extracell Vesicles. 2024.). To ensure standardization, we*
*maintained a consistent initial sample volume and normalized to Reads Per Million in our*
*sequencing analysis.*

17. The schematic diagram should distinguish between non-activated and activated mesothelial cells
for easy understanding.

*Thank you for your constructive and beneficial input. We've revised the schematic diagram to clearly*
*distinguish between non-activated and activated mesothelial cells for easier understanding.*

Reviewer #3 (Remarks to the Author):

In this study, the authors establish a correlation between the PROC and the JAK-STAT pathway

using bulk RNA sequencing. These results (i.e., upregulation of the JAK-STAT family) were then
independently confirmed in patient-derived samples through immunohistochemistry. The authors
then tested the effect of JAK pathway inhibition ex vivo and in vivo in well-established mouse
models using a known drug. Remarkably, the research group isolated extracellular vesicles (EVs)
emanating from ascites and cancer tissues. Through sequencing, they identified two particular
miRNAs that were enriched in the EVs from the resistant groups. They independently demonstrated
that transfection of these two miRNAs (independent of EV transfer) contributed to cisplatin
resistance. Using EVs enriched for these miRNA species, they showed that EV exposure to the
cancer cell model decreased sensitivity to cisplatin.

The effect of cisplatin sensitivity related to the two proposed miRNAs is consistent with the known
roles of related miRNAs (reviewed for instance in Yingchun Shao, Shuangshuang Zhang, Yuxin Pan,
Zhan Peng, Yinying Dong, miR-135b: A key role in cancer biology and therapeutic targets, Non-
coding RNA Research, Volume 12,2025). Importantly, the study integrates ex vivo and in vivo
validation and establishes a link between tumor-derived EVs and platinum resistance, adding to the
growing body of literature on the role of EVs in oncology. The study appears technically sound to
this reviewer.

The only minor concern is the need to formally establish whether the two enriched miRNAs are truly
encapsulated within the EVs or associated with the surface. It was not sufficiently clear in the
methods section whether this possibility was tested. Establishing formally the EV-mediated delivery
would be ideal but remain challenging. As this later point remains controversial, perhaps this should
be clearly added in the paragraph already highlighting the limitation of the study.

*We wish to express our deep appreciation to the reviewer for their insightful comment on this point.*
*To determine whether the two enriched miRNAs are truly encapsulated within the EVs or merely*
*associated with their surface, we conducted an additional experiment based on a previous study*
*(Vora A, et al. Proc Natl Acad Sci U S A. 2018.). The results of this additional work, suggested by the*
*reviewer, are presented in Fig. 6h and Supplementary Fig. 6h.*

*Methods: RNase and Triton X-100 treatment*

*The collected EVs were divided into three equal volumes: a negative control; RNase only; and Triton*
*X-100 and RNase. EVs were treated with or without 0.1% Triton X-100 (Thermo Fisher Scientific)*
*for 30 min at room temperature, followed by treatment with RNase A (5 µg/mL, Nippon gene) for 15*
*min at 37 °C. After RNase and/or Triton X-100 treatment, RNA extraction and quantitative reverse*
*transcription polymerase chain reaction were performed.*

*As the reviewer mentioned, miR-135 is consistent with the known roles of related miRNAs, but*
*demonstrating EV-mediated delivery remains a challenge. We have added the following text to the*
*Discussion.*

*Line 282–286: “we did not conduct a comprehensive analysis of the functions of EVs derived from*
*all cells in the ascites. EVs released by various unexamined cell types may play a role in regulating*
*the intraperitoneal cancer microenvironment and could contribute to platinum resistance. Ideally, we*
*would formally establish the mechanisms of EV-mediated delivery, but this remains a significant*
*challenge, representing a limitation of our study.”*